# GENIE: A Visual-Only Diffusion Framework for Task-Agnostic Image Transformation

**Uddeshya Singh**\*  *ud.uddeshya16@gmail.com*
*Center for Machine Intelligence and Data Science*
*Indian Institute of Technology Bombay*

**Aniket Thomas**\*  *aniket.thomas@iitb.ac.in*
*Center for Machine Intelligence and Data Science*
*Indian Institute of Technology Bombay*

**Aishwarya Agarwal**  *aishagar@adobe.com*
*Adobe Research, Bengaluru, India*

**Srikrishna Karanam**  *skaranam@adobe.com*
*Adobe Research, Bengaluru, India*

**Biplab Banerjee**  *getbiplab@gmail.com*
*Center for Machine Intelligence and Data Science*
*Indian Institute of Technology Bombay*

**Reviewed on OpenReview:** *https://openreview.net/forum?id=vtth9hOwoP*

## Abstract

Designing a unified vision model capable of handling diverse visual transformation tasks without task-specific modifications remains a significant challenge, particularly in scaling and generalizing beyond narrowly defined objectives. We propose **GENIE**[1], a novel *ControlNet-Diffusion* framework that performs task-based image generation solely through **visual exemplars**, eliminating dependence on textual prompts or auxiliary metadata. Unlike conventional prompt-driven diffusion models, GENIE employs a **dual visual conditioning** mechanism—combining implicit guidance via ControlNet and explicit task encoding through CLIP-based visual arithmetic—to infer task intent directly from reference input-output pairs. To improve semantic alignment between visual exemplars and generated outputs, we introduce a lightweight **task consistency loss**, which encourages representational coherence in the embedding space across transformed pairs. While not a multitask learner in the classical sense, GENIE employs a task-agnostic architecture that enables task switching across multiple image-to-image transformations without any task-specific modifications to the model architecture or loss functions. Instead of being explicitly provided with task identifiers, the model infers the intended task implicitly from the reference input–output pair through visual conditioning. Evaluations across **seven vision tasks**—inpainting, colorization, edge detection, deblurring, denoising, segmentation and depth estimation—and **four out-of-distribution (OOD) tasks on OOD data**—deraining, contrast enhancement , map to aerial and scribble to image generation—demonstrate that GENIE achieves an average performance gain of **7.13%** over other visual prompt conditioned baselines, showcasing its effectiveness for scalable and text-free visual generation.

---

\* Equal contribution.
[1]https://github.com/AniTho/Visual-Prompting-GENIE

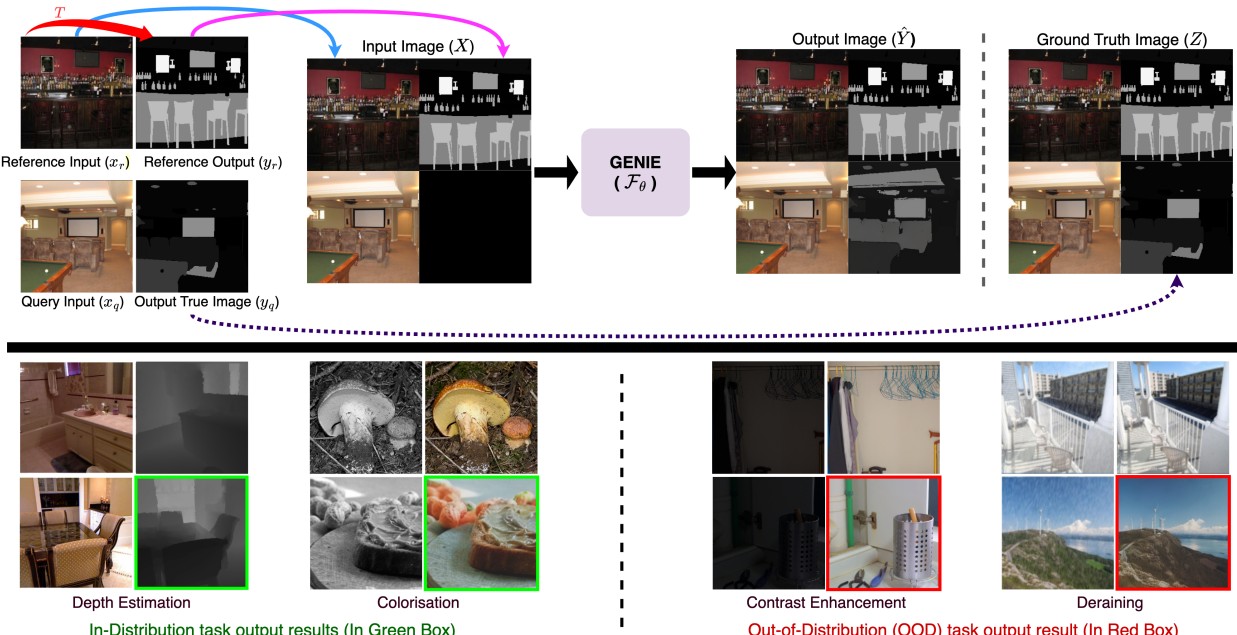

Figure 1: **Illustration of the proposed model, GENIE** ($\mathcal{F}_\theta$), performing diverse visual tasks solely through visual guidance. The **top section** illustrates the model's workflow, where a $2 \times 2$ input grid $X$ contains reference input-output pairs $(x_r, y_r)$, a query input image $(x_q)$, and a masked region. GENIE generates a predicted output $\hat{Y}$, shown alongside the ground truth $Z$; the bottom-right cell represents the target output $(y_q)$. The **bottom section** demonstrates GENIE's predictions across various tasks, highlighting its effectiveness on both in-distribution tasks (green boxes) and out-of-distribution (OOD) tasks on OOD data (red boxes).

# 1 Introduction

The field of computer vision has seen tremendous advancements through specialized models targeting isolated tasks like classification He et al. (2016); Dosovitskiy et al. (2021); Huang et al. (2017); Tan & Le (2019) and segmentation (Kirillov et al., 2023; Chen et al., 2018; Ronneberger et al., 2015). Despite their high performance, these models are typically architecturally rigid and limited in their ability to generalize across tasks or domains without retraining. As a result, scalability and adaptability remain critical bottlenecks when attempting to build general-purpose visual systems.

In contrast, *in-context learning* (ICL) (Brown et al., 2020) has transformed natural language processing (NLP) by enabling models to perform new tasks at inference time through conditioning on a few input-output examples, without retraining. This strategy, successful in zero-shot and few-shot NLP (Chowdhery et al., 2023; Du et al., 2022; Wei et al., 2022), has inspired growing interest in extending ICL principles to computer vision.

However, translating ICL to vision is not straightforward, and recent attempts fall into two distinct categories. *Multi-modal approaches* integrate vision and language, relying on textual prompts to specify the desired task (Alayrac et al., 2022; Lu et al., 2023; Wang et al., 2023a; Geng et al., 2024). While powerful, they are inherently limited in settings where text is unavailable, ambiguous, or insufficient to capture complex visual intent. *Vision-only models* circumvent this by using grids of visual examples to condition the model purely through input-output image pairs (Bar et al., 2022; Wang et al., 2023b; Bai et al., 2024) (Fig.1). These models often lack mechanisms for explicit task-specific alignment, leading to ambiguity, reduced output fidelity, and poor generalization when faced with complex or out-of-distribution (OOD) tasks (Fig.2).

A representative failure case of text-reliant models is shown in Fig.3, where a vision and text-conditioned model (Wang et al., 2023c; Meng et al., 2024) might fail to infer the intended transformation when textual

prompt is removed and generate inconsistent result with only visual references. This highlights the need for robust and vision-centric alternatives that can infer intent directly from visual cues.

To address these limitations, we introduce **GENIE** (***Ge**nerative **N**etwork using **I**mage **E**xemplars*), a diffusion-based, text-free framework for exemplar-guided image transformation. GENIE is trained jointly on multiple tasks as a single system with one backbone, one output head, and a lightweight loss function. At inference time, it seamlessly switches between transformations by ingesting different input-output exemplar pairs, demonstrating promising zero-shot generalization to new tasks and domains without any architectural or loss-function modifications.

At the core of GENIE are three key technical components designed to tackle the challenges in exemplar-guided image transformation: **(a)** A *dual visual conditioning mechanism* that combines implicit spatial features from ControlNet (implicit conditioning) with explicit task representations from CLIP-based (Radford et al., 2021) visual arithmetic (explicit conditioning). This directly addresses the task alignment problem seen in prior vision-only models and also removes textual dependency from diffusion models. **(b)** A *masked image modeling objective*, which allows generalization and allows the single architecture to support a wide variety of transformation tasks without needing output-space redesigns for each task. **(c)** A lightweight yet effective *task consistency loss* that operates in CLIP's embedding space to enforce semantic coherence between the reference and predicted transformations, ensuring the model preserves transformation intent, especially under domain shifts.

While both InstructGIE (Meng et al., 2024) and Prompt Diffusion (Wang et al., 2023c) use a reference–query grid (akin to our implicit conditioning), our explicit conditioning goes beyond this by replacing text with CLIP-based visual arithmetic to inject the transformation from reference to query and thereby replacing text conditioning, making entire architecture purely visually conditioned. This is made effective by our task consistency loss, which ensures that the transformation vectors of the reference and query, captured by visual arithmetic remain aligned and transfers task intent effectively.

Collectively, these components enable GENIE to generalize effectively across task boundaries, outperforming chosen only-vision prompt conditioned baselines—including Painter (Wang et al., 2023b), Visual Prompt (Bar et al., 2022), and LVM (Bai et al., 2024). While numerous related methods exist, these mostly fall into two categories—masked image modeling (MIM) and autoregressive next-token prediction (NTP). We select three representative proxies: Painter for MIM, LVM for autoregressive NTP, and Visual Prompt for MIM with self-supervision on open-set tasks. These baselines are strictly vision-only and not compared against architectures with textual inputs.

Quantitative experiments conducted across seven in-distribution (ID) and four out-of-distribution (OOD) tasks on OOD data reveal that GENIE achieves a consistent **7.13% average performance improvement across all tasks, measured relative to the second-best performing model for each task**, all without text, retraining, or task-specific modules.In addition, qualitative results on ID tasks evaluated using OOD data further demonstrate the model's ability to handle tasks beyond the dataset it was trained on,

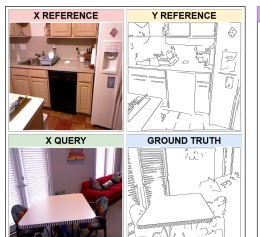 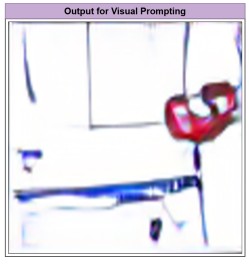 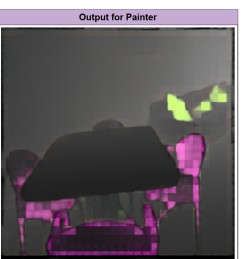 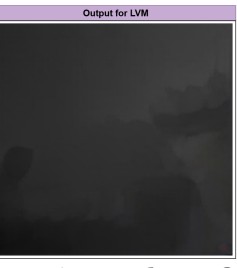 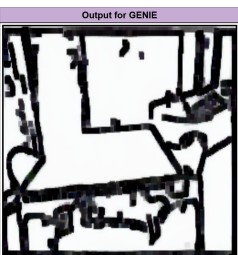

Figure 2: **Comparison of outputs on the OOD scribble conversion task on OOD Data:** (a) Input example with both query and reference images; (b) Output from Visual Prompt (Bar et al., 2022), showing limited task adaptation; (c) Output from Painter (Wang et al., 2023b), struggling to interpret the task; (d) Output from LVM (Bai et al., 2024), also failing to interpret the task effectively; (e) Output from GENIE (ours), demonstrating robust and effective handling of the OOD task on OOD Data. *All models were retrained under the same conditions for fair comparison.*

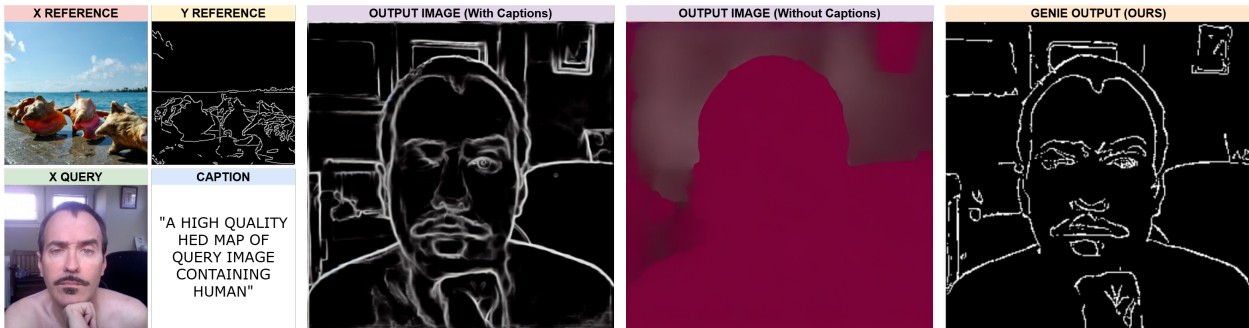

Figure 3: **GENIE vs. Prompt-Diffusion under prompt ablation:** (a) Input exemplar grid with reference and query samples; (b) Output from Prompt-Diffusion using a caption prompt; (c) Output from Prompt-Diffusion without a caption prompt, showing a significant drop in performance due to prompt dependence; (d) Output from GENIE (ours), which performs robustly without requiring textual input, demonstrating strong generalization using purely visual guidance.

| Model | Textual Conditioning | Visual Conditioning | External Visual Conditioning | Reconstruction Loss | Task Constraint Loss |
|---|---|---|---|---|---|
| Visual Prompt (Bar et al., 2022) | ✗ | ✓ | ✗ | ✓ | ✗ |
| Unified-IO (Lu et al., 2023) | ✓ | ✓ | ✗ | ✓ | ✗ |
| Painter (Wang et al., 2023b) | ✗ | ✓ | ✗ | ✓ | ✗ |
| Prompt Diffusion (Wang et al., 2023c) | ✓ | ✓ | ✗ | ✓ | ✗ |
| LVM (Bai et al., 2024) | ✗ | ✓ | ✗ | ✓ | ✗ |
| PromptGIP (Liu et al., 2024b) | ✗ | ✓ | ✗ | ✓ | ✗ |
| InstructGIE (Meng et al., 2024) | ✓ | ✓ | ✗ | ✓ | ✓ |
| **GENIE (ours)** | ✗ | ✓ | ✓ | ✓ | ✓ |

Table 1: **Comparison of visual conditioning models** across key architectural attributes. GENIE uniquely combines external visual guidance and task-specific loss without textual prompts.

highlighting its generalization capability. Our model's distinct advantages are further detailed in Table 1. In summary, our contributions are:

- We propose **GENIE**, a unified, text-free image transformation framework based on diffusion model that integrates both *implicit* (ControlNet-based spatial guidance) and *explicit* (CLIP-based visual feature arithmetic) conditioning to infer task intent directly from images.

- We design a **task consistency loss** that operates in CLIP embedding space to enforce semantic alignment between reference and query transformations, enabling task adherence even under domain and task distribution shifts.

- We demonstrate the effectiveness of GENIE across seven diverse ID tasks and four OOD tasks on OOD data, showing up to **7.3% performance gains** over state-of-the-art only visual conditioned prompting methods.

## 2 Related Works

**Masked Image Modeling.** Masked Image Modeling is a core strategy in self-supervised learning. Methods like Context Encoders (Pathak et al., 2016) and MAE (He et al., 2022) learn powerful, general-purpose visual representations by reconstructing masked image patches. While effective for pretraining, these models are not designed to infer and perform tasks from visual exemplars directly.

**In-Context Learning (ICL) with Diffusion Models.** ICL, widely used in NLP (Brown et al., 2020; Radford et al., 2019), multimodal extensions such as Flamingo (Alayrac et al., 2022), Unified-IO (Lu et al.,

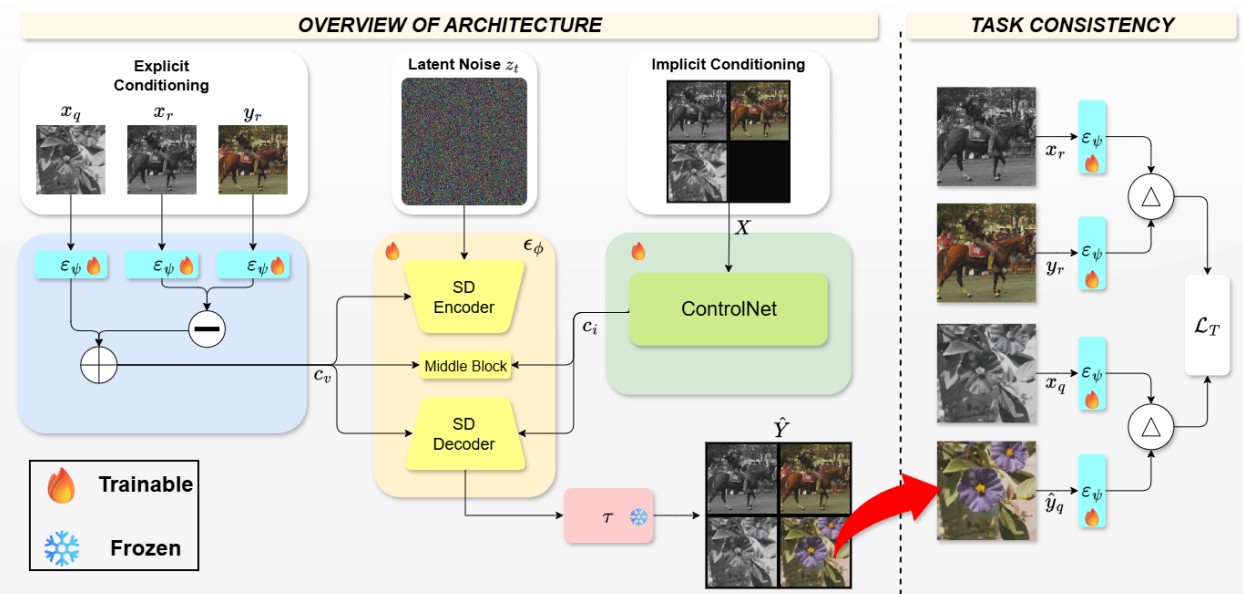

Figure 4: **Overview of the GENIE architecture.** The **left section** illustrates GENIE's dual visual conditioning framework. A $2 \times 2$ image grid ($X$), comprising a query input ($x_q$), reference input ($x_r$), and reference output ($y_r$), is processed through: (i) *explicit conditioning*, where a CLIP-based encoder ($\varepsilon_\psi$) computes a visual task representation ($c_v$) via visual arithmetic, and (ii) *implicit conditioning*, where ControlNet extracts spatial cues ($c_i$) directly from $X$. These embeddings jointly condition the latent diffusion model ($\varepsilon_\phi$) to generate the query output ($\hat{y}_q$). The **right section** depicts the *task consistency* module, wherein the CLIP encoder ($\varepsilon_\psi$) encodes $x_r$, $y_r$, $x_q$, and $\hat{y}_q$ to enforce semantic consistency between the reference and generated transformations through the task consistency loss ($\mathcal{L}_T$).

2023), etc. In the visual generative domain, diffusion models (Ho et al., 2020; Rombach et al., 2022) are state-of-the-art. ControlNet (Zhang et al., 2023) introduced strong visual conditioning but requires separate modules for each task. Subsequent works like Prompt Diffusion (Wang et al., 2023c), InstructDiffusion (Geng et al., 2024) and InstructGIE (Meng et al., 2024) achieve task generalization by fusing visual and textual prompts. However, their fundamental dependence on text limits their use in scenarios that are purely visual or where text is ambiguous.

**Visual Prompting.** To eliminate text dependency, text-free visual prompting methods like PromptGIP (Liu et al., 2024b), Visual Prompt (Bar et al., 2022) and Painter (Wang et al., 2023b) condition models on masked input-output image grids, following BERT (Devlin et al., 2019) style training. These approaches rely on token-level self-attention to infer the task. While effective for some in-distribution tasks, this lack of explicit task alignment can lead to poor generalization on fine-grained or OOD transformations.

**Large Vision Models.** A recent class of Large Vision Models (Bai et al., 2024; Hao et al., 2024) follows a vision-only, autoregressive paradigm, similar to language models predicting the next token. These models show impressive generality but do not incorporate external conditioning, which may limit their effectiveness on tasks that benefit from structured visual guidance.

In summary, current approaches either rely on textual prompts or, if purely visual, lack explicit task grounding, leading to ambiguity in complex or OOD settings. This gap motivates our proposed framework: a text-free diffusion model with dual visual conditioning and a task consistency loss for robust, exemplar-guided transformation.

# 3 Proposed Methodology

We propose **GENIE**, a visual-only diffusion framework for prompt-guided conditional generation that performs multiple tasks without any task-specific modifications, using only visual examples without textual prompts or task labels. Its key novelty is a *dual visual conditioning* mechanism that combines implicit token-level guidance through self-attention and explicit semantic guidance via vector arithmetic, conditioned through cross-attention, which replaces the need for textual prompts. Additionally, our novel *task consistency loss* enforces that the semantic transformation from reference input to output is preserved in the query input to its generated output by aligning their embedding. Detailed discussions follow.

## 3.1 Problem Formulation

Let $(x_r, y_r)$ denote a reference input–output pair, where $y_r = T(x_r)$ and $T$ is an unknown transformation (e.g., colorization, edge detection, or depth estimation). Given a query input $x_q$, the goal is to generate an output $\hat{y}_q$ such that $\hat{y}_q \approx T(x_q)$, purely by inferring $T$ from the exemplar $(x_r, y_r)$. The true output is denoted as $y_q = T(x_q)$. All images are of shape $\mathbb{R}^{H \times W \times 3}$, with $H = W = 256$.

We arrange the inputs into a $2 \times 2$ spatial grid forming a composite image $X \in \mathbb{R}^{2H \times 2W \times 3}$, posing the task as image completion where the model fills in the masked region for $y_q$, following the setup in (Wang et al., 2023b; Bar et al., 2022) as is illustrated in Fig. 4.

$$X = \begin{bmatrix} x_r & y_r \\ x_q & [\text{MASK}] \end{bmatrix}$$

During training, a corresponding completed version $Z$ is constructed by populating the masked region with $y_q$. The model $\mathcal{F}_\theta$ is trained to reconstruct $Z$ from $X$ and a noisy latent $z_t$, sampled for timestep $t$ of the denoising diffusion process, with a focus on generating the missing query output:

$$\hat{y}_q = \mathcal{F}_\theta(z_t, X, x_r, y_r, x_q), \quad \text{and} \quad y_q = T(x_q)$$

Unlike models such as Painter (Wang et al., 2023b), our framework normalizes all images to $[0, 1]$ across tasks, avoiding any data-specific modifications and enabling seamless integration of heterogeneous transformations.

## 3.2 Architecture Overview

As shown in Fig. 4, GENIE is built upon the latent diffusion model (Rombach et al., 2022), augmented with a ControlNet module (Zhang et al., 2023) for implicit visual guidance and a dedicated CLIP-based (Radford et al., 2021) visual encoder for explicit visual guidance.

Given an unmasked $2 \times 2$ grid $Z \in \mathbb{R}^{512 \times 512 \times 3}$, GENIE follows the latent diffusion paradigm (Rombach et al., 2022) by first encoding the input into a compact latent representation using a pre-trained VQ-GAN (Esser et al., 2021) encoder $\tau_{enc}$. This latent compression enables efficient training and inference in a reduced-dimensional space while preserving essential visual structure. The resulting latent representation is defined as

$$z_0 = \tau_{enc}(Z), \tag{1}$$

where the corresponding VQ-GAN decoder $\tau_{dec}$ is later used to reconstruct the output from the denoised latent. Together, the encoder and decoder are denoted as $\tau$.

Although VQ-GAN is typically pre-trained on natural single images, we use it to encode and decode concatenated $2 \times 2$ image grids. To verify that this does not introduce significant reconstruction artifacts due to distribution shift, we quantitatively evaluate reconstruction fidelity on grid inputs. Using the pre-trained VQ-GAN without fine-tuning, we observe a mean L1 reconstruction loss of **0.033** and a mean RMSE of **0.073** across 10,000 samples, indicating that the grid structure is preserved. In addition, Fig. 5 provides a qualitative comparison between the original grid inputs and their reconstructed outputs, further demonstrating that visual content and spatial structure are well preserved. These results suggest that the pre-trained

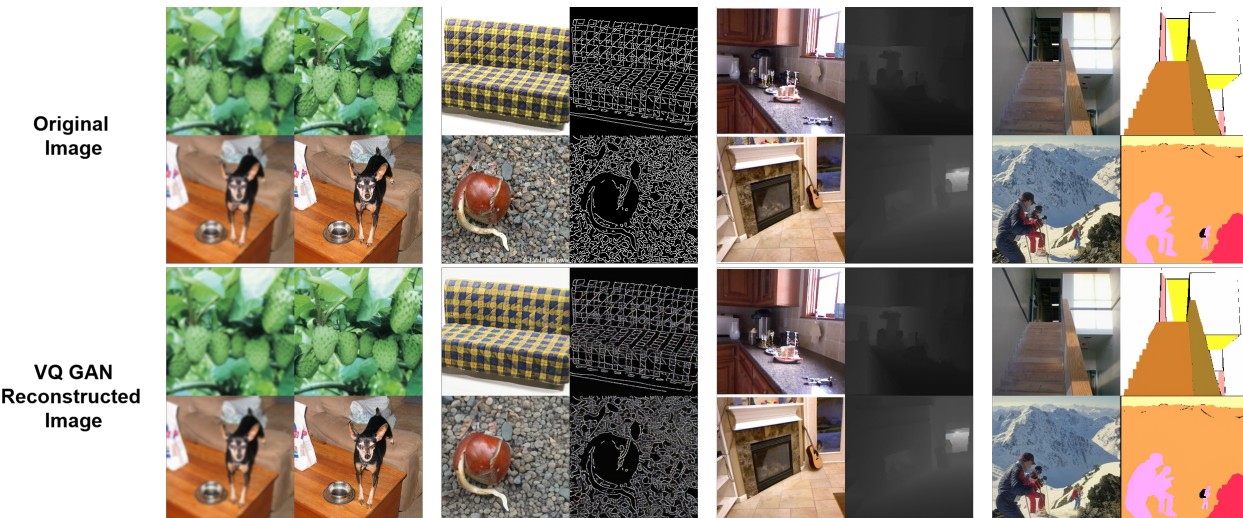

Figure 5: Qualitative reconstruction results of the pre-trained VQ-GAN on concatenated $2 \times 2$ image grids. The original input grids (top row) and their corresponding reconstructed outputs after VQ-GAN encoding and decoding (bottom row) show that spatial structure and visual content are well preserved despite the use of a VQ-GAN pre-trained on single natural images.

VQ-GAN generalizes sufficiently to tiled grid inputs and does not adversely affect downstream generation quality.

To guide the generation process in diffusion, models are commonly conditioned on textual prompts using classifier-free guidance (Ho & Salimans, 2022a). To make our model text-free and rely solely on visual cues, we instead utilize explicit conditioning ($c_v$) via a CLIP (Radford et al., 2021) vision encoder $\varepsilon_\psi$ which maps $x_r$, $y_r$, and $x_q$ to 1024-D CLIP embeddings, and uses task vector arithmetic to construct $c_v$, which represents the intended transformation direction and serves as a semantic prior.

In addition to explicit semantic conditioning, GENIE utilizes a ControlNet conditioning module to extract spatially aligned implicit conditioning $c_i$ via self-attention from the masked composite grid $X$. While the explicit condition captures the intended transformation in a semantic embedding space, the implicit condition provides spatially aligned structural cues from the input, enabling the model to jointly leverage semantic intent and spatial context during the diffusion process. A more detailed formulation of the dual visual conditioning mechanism is provided in Section 3.3.

$$e_{x_r} = \varepsilon_\psi(x_r), \quad e_{y_r} = \varepsilon_\psi(y_r), \quad e_{x_q} = \varepsilon_\psi(x_q) \tag{2}$$

$$c_v = e_{y_r} - e_{x_r} + e_{x_q} \quad \text{(explicit visual condition)} \tag{3}$$

$$c_i = \text{ControlNet}(X) \quad \text{(implicit condition)} \tag{4}$$

Starting from the compressed latent representation $z_0 \in \mathbb{R}^{64 \times 64 \times 4}$, GENIE follows the standard diffusion process by progressively adding noise over $t$ timesteps to obtain the noisy latent $z_t$. A Transformer-U-Net denoising network $\epsilon_\phi$ then predicts the noise component at each timestep, conditioned jointly on the diffusion timestep $t$, the explicit semantic condition $c_v$, and the implicit spatial condition $c_i$:

$$\hat{\epsilon} = \epsilon_\phi(z_t, t, c_v, c_i). \tag{5}$$

Each denoising block integrates convolutional layers, ResNet modules (He et al., 2016), and Vision Transformer components (Dosovitskiy et al., 2021), allowing the network to capture both local visual details and global contextual information.

After denoising, the predicted noise $\hat{\epsilon}$ is subtracted from the noisy latent $z_t$ to recover the denoised latent $\tilde{z}_0$, which is then passed through the VQ-GAN decoder to reconstruct the full grid:

$$\tilde{z}_0 = z_t - \hat{\epsilon}. \tag{6}$$

$$\hat{Y} = \tau_{dec}(\tilde{z}_0). \tag{7}$$

Finally, the bottom-right (BR) quadrant of this reconstructed grid, corresponding to the query output, is extracted as

$$\hat{y}_q = \hat{Y}^{\mathrm{BR}}. \tag{8}$$

All components are combined in a unified model $\mathcal{F}_\theta$ and trained end-to-end, enabling the model to jointly preserve structural fidelity through implicit conditioning and semantic alignment through explicit visual guidance.

### 3.3 Dual Visual Conditioning

**(a) Implicit Conditioning via ControlNet.** The composite image $X$, containing the reference pair $(x_r, y_r)$ and the query input $x_q$, is fed into a ControlNet module. Through self-attention, the model extracts implicit features $c_i$ that capture task-relevant visual transformations such as color changes, structural modifications, or denoising patterns, and adapts them to the query content. These features are injected into multiple layers of the denoising network, providing fine-grained spatial cues without any explicit task labels. The entire conditioning process is trained end-to-end to complete the masked region in $X$.

Since $X$ implicitly defines the transformation through exemplar placement, this structure allows GENIE to generalize across task types. While the current approach samples $(x_r, y_r)$ randomly, future work can incorporate selection strategies based on visual or semantic similarity.

**(b) Explicit Conditioning via Visual Arithmetic.** To capture high-level task semantics, we utilize a CLIP ViT-B/32 encoder $\varepsilon_\psi(\cdot)$ to extract 1024-D semantic embeddings for $x_r$, $y_r$ and $x_q$. Compared to alternatives like VQ-VAE (Rombach et al., 2022), ViTs (Dosovitskiy et al., 2021), or LLaVA-style caption generators (Liu et al., 2024a), CLIP embeddings were found to offer the best trade-off between task-discriminability and image content (see Sup Mat).

$$e_{x_r} = \varepsilon_\psi(x_r), \quad e_{y_r} = \varepsilon_\psi(y_r), \quad e_{x_q} = \varepsilon_\psi(x_q)$$

We compute the reference task vector via:

$$v_r = e_{y_r} - e_{x_r} \tag{9}$$

This difference vector represents the semantic shift induced by the transformation $T$. To apply $T$ to $x_q$, we generate a target embedding via:

$$c_v = e_{x_q} + v_r \tag{10}$$

This analogy-inspired construction draws from the insight that CLIP's embedding space $\mathcal{E}$ encodes meaningful transformations as linear paths (Radford et al., 2021; Reed et al., 2015). If $T$ maps $x_r \rightarrow y_r$, then $c_v$ approximates the desired location of $\hat{y}_q$ in $\mathcal{E}$.

During denoising, $c_v$ is injected into all cross-attention layers of the denoising network, which helps to disambiguate tasks that have visually similar structures but distinct semantics, and removes the need for text conditioning.

During training, we randomly drop the explicit semantic condition $c_v$ with probability 0.5. This strategy is inspired by ControlNet (Zhang et al., 2023), dropping $c_v$ encourages GENIE to infer the task solely from the implicit conditioning $c_i$ and helps the model towards classifier-free guidance (Ho & Salimans, 2022b;a).

### 3.4 Learning Objective

GENIE is trained using a composite loss that combines generative fidelity with semantic alignment.

**(a) Diffusion Loss.** We adopt the standard latent denoising objective from diffusion models (Rombach et al., 2022). Let $z_0$ denote the clean latent encoding of the composite grid $Z$, and $z_t$ its noisy version at timestep $t$. The denoising U-Net $\epsilon_\phi$ predicts the added noise $\epsilon$, conditioned on both implicit ($c_i$) and explicit ($c_v$) visual cues:

$$\mathcal{L}_D = \mathbb{E}_{z_0,t,\epsilon \sim \mathcal{N}(0,1)} \left[ \|\epsilon - \epsilon_\phi(z_t, t, c_v, c_i)\|_2^2 \right] \tag{11}$$

**(b) Task Consistency Loss.** To ensure that the model performs the correct semantic transformation on the query image based solely on the reference pair, we propose a task consistency loss $\mathcal{L}_T$ that promotes structural equivalence of transformations in the embedding space. Rather than relying on explicit task labels, we assume that transformations manifest as vector displacements in a shared semantic manifold $\mathcal{E} \subset \mathbb{R}^d$, parameterized by a CLIP-based visual encoder $\varepsilon_\psi$.

Inspired by analogy-based learning (Reed et al., 2015; Hertzmann et al., 2023), we model a transformation $T$ as the difference between the reference pair embeddings:

$$v_r = \varepsilon_\psi(y_r) - \varepsilon_\psi(x_r). \tag{12}$$

After conditioning the diffusion model on the reference pair and query input $x_q$, the output $\hat{y}_q$ is reconstructed by decoding the denoised latent:

$$\hat{y}_q = \tau \left( z_t - \epsilon_\phi(z_t, t, c_v, c_i) \right), \tag{13}$$

and the query transformation is similarly computed as:

$$v_q = \varepsilon_\psi(\hat{y}_q) - \varepsilon_\psi(x_q). \tag{14}$$

To enforce semantic consistency, we define the task consistency loss by matching the *L2 norm* of the difference between the transformation vectors:

$$\mathcal{L}_T = \mathbb{E} \left[ \|v_r - v_q\|_2^2 \right]. \tag{15}$$

**Further Insights to $\mathcal{L}_T$:** From an analytical point of view, the loss operates in a feature space structured by contrastive pretraining of $\varepsilon_\psi$ where semantically meaningful concepts are organized as direction. t-SNE clustering in the Sup Mat supports this assumption, showing that input–output difference vectors, indicating the direction from input to output, naturally group by task.

The task consistency loss, based on above explanation, is designed to ensure that the semantic transformation captured by the reference pair is similar to the semantic transformation between the query input and its generated output. It does not assume uniformity across all tasks (e.g., deraining under light vs. heavy rain) but instead ensures that the relative semantic shift observed in the reference pair is mirrored in the query. This is done by minimizing the difference between their task representation (Eqn. 12 and Eqn. 14) in embedding space, providing a lightweight inductive bias that stabilizes the strength of transformations inferred from exemplars. Thus encouraging both pairs to represent the same task strength without requiring identical pixel-level changes. Geometrically, it enforces that $v_r$ and $v_q$ lie on the same hypersphere in $\mathcal{E}$, preserving the geodesic radius of the transformation. Operating in a contrastively structured feature space, this avoids both under- and over-editing, thereby supporting generalization in both ID and OOD settings where task style may vary but transformation strength should remain coherent.

**Final Objective.** GENIE is trained to using:

$$\mathcal{L} = \mathcal{L}_D + \lambda_T \cdot \mathcal{L}_T \tag{16}$$

We empirically set $\lambda_T = 0.05$ to balance generation quality and transformation alignment. This weighting allows GENIE to consistently perform high-fidelity synthesis while preserving the intended transformation

| Category | Dataset | Associated Tasks |
|---|---|---|
| In-Distribution (ID) Data & Task | ImageNet (Deng et al., 2009) | Deblurring, Super-resolution, Inpainting, Colorization, Edge Detection, Denoising |
| | ADE20K (Zhou et al., 2017) | Semantic Segmentation |
| | NYU Depth V2 (Silberman et al., 2012) | Depth Estimation |
| Out-of-Distribution (OOD) Data & Task | Deraining (Zamir et al., 2022) | Deraining |
| | LOL (Wei et al., 2018) | Low-light Enhancement |
| | Map-to-Aerial (Isola et al., 2017) | Image Generation |
| | Scribble-to-Image (Isola et al., 2017) | Image Generation |
| OOD Data & ID Task | DomainNet (Peng et al., 2019) | Denoising, Inpainting, Edge Detection, Colorization |

Table 2: **Overview of dataset–task configurations** used for evaluation. OOD settings—either due to novel tasks or unseen domains—are highlighted in gray.

semantics across a wide range of tasks. Increasing $\lambda_T$ further improves task alignment but leads to visual artifacts, revealing a trade-off between semantic fidelity and perceptual quality. This choice is also motivated by findings in prior works (Isola et al., 2017), where a high weighting ($\lambda = 100$) is placed on reconstruction objectives to emphasize alignment.

## 4 Experimental Evaluations

### 4.1 Training Setup and Dataset Details

For each task, we curated a diverse set of 20,000 randomly sampled images from multiple datasets (detailed in Table 2), to support a range of vision tasks. ImageNet (Deng et al., 2009), ADE20K (Zhou et al., 2017), and NYU Depth V2 (Silberman et al., 2012) were used as primary sources to ensure visual diversity and broad task coverage. To assess GENIE's generalization, we structure the evaluation into three categories (Table 2): (a) in-distribution (ID) tasks on ID data (e.g., denoising), (b) out-of-distribution (OOD) tasks from OOD data (e.g., deraining), and (c) ID tasks evaluated on visually dissimilar OOD domains using DomainNet (Peng et al., 2019). Additional dataset details are in the Sup Mat.

All components of the GENIE are jointly trained end-to-end on all tasks, without any task specific modification to architecture or loss function, using AdamW optimizer with a weight decay of 0.01 (Loshchilov & Hutter, 2019) and a learning rate of $1 \times 10^{-5}$ on a single NVIDIA A100 GPU with 80GB of memory, using a batch size of 16 for 20 days, ensuring fairness in comparison and evaluation.

### 4.2 Evaluation Strategy

We evaluate deblurring, super-resolution, denoising, colorization, and inpainting on the ImageNet validation set (Deng et al., 2009) using task-specific metrics: PSNR for deblurring and super-resolution; PSNR and SSIM for denoising; MSE for colorization; and SSIM and MSE for inpainting. Depth estimation is evaluated using RMSE on the NYU Depth V2 validation set (Silberman et al., 2012). For semantic segmentation, we evaluate on the ADE20K validation set (Zhou et al., 2017) using Adjusted Rand Index (ARI) which is a clustering-based, label-agnostic metric to assess the structural consistency of the generated segmentations. We also report LPIPS (Zhang et al., 2018) for all image reconstruction tasks to assess perceptual quality.

To assess GENIE's generalization to OOD tasks, we evaluate its performance on deraining using the dataset (Zamir et al., 2022), on contrast enhancement using the LOL dataset (Wei et al., 2018), using PSNR, SSIM and LPIPS, and on image generation from the map-to-aerial and scribble-to-image subset of pix-2-pix dataset (Isola et al., 2017), using LPIPS. We further conduct qualitative analysis on the Painting, Sketch, and Clipart subsets of DomainNet (Peng et al., 2019) to examine robustness under domain shift.

GENIE is compared against three vision-only baselines: Visual Prompting (Bar et al., 2022), Painter (Wang et al., 2023b), and LVM (Bai et al., 2024). All models are retrained and evaluated across a shared set of tasks as described above to ensure fair and consistent comparison.

| | Depth Estimation | Denoising | | | Super Resolution | | Deblurring | | Colorization | | Inpainting | | | Semantic Segmentation |
|---|---|---|---|---|---|---|---|---|---|---|---|---|---|---|
| **In Distribution Task** | | | | | | | | | | | | | | |
| Models | RMSE ↓ | PSNR ↑ | SSIM ↑ | LPIPS ↓ | PSNR ↑ | LPIPS ↓ | PSNR ↑ | LPIPS ↓ | MSE ↓ | LPIPS ↓ | SSIM ↑ | MSE ↓ | LPIPS ↓ | ARI ↑ |
| Visual Prompt (Bar et al., 2022) | 0.112 | 17.513 | 0.532 | 0.411 | 17.786 | 0.397 | 18.020 | 0.384 | 0.028 | 0.337 | 0.374 | 0.044 | 0.557 | 0.303 |
| Painter (Wang et al., 2023b) | 0.063 | 19.060 | **0.842** | 0.366 | 19.230 | 0.386 | **24.331** | **0.161** | 0.022 | 0.264 | 0.429 | 0.030 | 0.442 | **0.65** |
| LVM (Bai et al., 2024) | 0.071 | 19.413 | 0.793 | 0.373 | 18.687 | 0.372 | 22.246 | 0.377 | 0.019 | 0.261 | 0.558 | 0.018 | 0.411 | 0.573 |
| GENIE | **0.061** | **23.712** | 0.827 | **0.261** | **22.323** | **0.271** | 23.210 | 0.218 | **0.016** | **0.255** | **0.632** | **0.013** | **0.381** | 0.613 |

Table 3: **Quantitative results of models on ID Tasks**. Metrics marked with "↓" indicate lower is better, and metrics marked with "↑" indicate higher is better. The hyphen symbol "-" indicates that the model outputs unrelated or ambiguous results, making quantitative evaluation unsuitable. Best results are in "bold" and second-best are underlined.

## 4.3 Qualitative Results

We qualitatively compare GENIE with Painter, Visual Prompting, and LVM on both in-distribution (green boxes) and OOD tasks (red boxes) in Fig. 6.

For ID tasks on ID data, Painter produces high-quality outputs. Visual Prompting correctly infers the task but often introduces structural artifacts. In contrast, LVM and GENIE consistently generate high-fidelity, semantically accurate outputs that preserve fine image details across all tasks. On OOD tasks on OOD data, Painter's performance degrades significantly; it fails to adapt and often reverts to identity mappings or familiar in-distribution behaviors. Visual Prompting again shows moderate task inference but produces distorted and blurred results.

GENIE and LVM generalize far better to these unseen tasks. However, their performance varies: LVM excels at deraining but struggles with contrast enhancement and scribble generation (Fig. 2). This is likely because deraining shares low-level priors (e.g., texture removal) with LVM's pretraining, whereas tasks like contrast enhancement require domain-specific color transformations that GENIE is better equipped to model. Notably, GENIE also visually outperforms LVM on in-distribution tasks when applied to OOD-style data on ID task (e.g., Painting and Clipart datasets). While both models show some degradation on OOD tasks, the outputs remain visually coherent, highlighting their potential as generalized models. More qualitative results for GENIE are available in the Sup Mat.

## 4.4 Quantitative Results

We quantitatively evaluated GENIE on a wide range of ID tasks on ID data and OOD tasks on OOD data, with results summarized in Table 3 and Table 4. Baselines that failed to generate valid outputs on certain tasks were excluded from comparison.

**In-Distribution Tasks on In-Distribution data**: GENIE performs strongly across ID tasks on ID data, achieving top performance in depth estimation, super-resolution, colorization, and inpainting. For *denoising*, both GENIE and LVM outperform Painter in PSNR, while showing slightly lower SSIM and LPIPS. This behavior is expected, as latent-space reconstruction prioritizes structural fidelity over perceptual smoothness, an effect also observed in deblurring. For *semantic segmentation*, although GENIE does not explicitly optimize for pixel-level class labels, it consistently captures object-level structure and boundaries across complex scenes (Fig. 6, Fig. 9, Fig. 10) and across domains (Fig. 7). As GENIE operates in a latent generative framework without dense supervision, it may occasionally assign incorrect class identities despite producing semantically coherent segmentations. This limitation is shared by other latent-space models such as LVM and Visual Prompting, and suggests that mIoU alone—being sensitive to exact label assignment—can underestimate generation quality. To provide a fairer assessment, we evaluate segmentation using the clustering-based Adjusted Rand Index (ARI), which captures whether the pixels belonging to the same object are grouped together, without penalizing incorrect class assignments. We adopt this metric for a more meaningful assessment of segmentation quality by the latent generative framework models when semantic labeling is imperfect. These results, reported in Table 3, show that GENIE produces structurally consistent segmentations comparable to other baselines. Overall, GENIE achieves an average improvement of **8.3%** over the second-best method across all ID tasks on ID data.

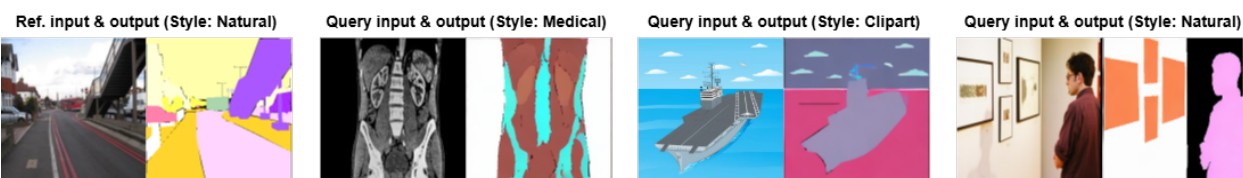

Figure 6: **Comparison of model results across various tasks.** The left portion shows results on in-distribution (green outline) and OOD tasks (red outline), while the right portion displays performance on a different domain data (Peng et al., 2019). While *Painter* (Wang et al., 2023b) struggles with OOD tasks and *Visual Prompt* (Bar et al., 2022) often distorts structures, both *LVM* (Bai et al., 2024) and *GENIE* show strong performance across many tasks. However, *GENIE* demonstrates superior generalization, producing better results than *LVM*. Also, *LVM*, *Visual Prompt*, and *GENIE* successfully segment objects but often misassign labels due to patch reconstruction in latent space.

Figure 7: The model maintains consistent segmentation quality across diverse input domains, including natural images, medical scans, and clipart-style scenes.

**Out-of-Distribution Tasks on OOD data**: GENIE demonstrates strong generalization on unseen out-of-distribution (OOD) tasks such as contrast enhancement and deraining, consistently outperforming the next-best baselines, Visual Prompting and LVM. On *contrast enhancement*, GENIE surpasses LVM by 12.6% in PSNR, 25.9% in SSIM, and 5.8% in LPIPS. For *deraining*, it achieves a 13.3% improvement in SSIM while maintaining competitive PSNR and LPIPS. We also quantitatively evaluate models on map-to-aerial and scribble-to-realistic tasks which inherently are image synthesis tasks. We noticed GENIE outperforming LVM by 8% in LPIPS and approximately 1.7% on both the tasks respectively. Overall, GENIE attains a mean improvement of **5.96%** over LVM across OOD tasks, highlighting its ability to infer and generalize transformations from exemplars beyond the training distribution. As Painter (Wang et al., 2023b) performs poorly on these OOD tasks and often fails to produce semantically meaningful outputs in the OOD task on OOD data setting(see Fig. 6). This makes quantitative metrics misleading.

In total, GENIE achieves an average performance gain of approximately **7.13%** across all evaluated tasks, validating its effectiveness as a unified framework for diverse image-to-image translation.

### 4.5 Reference Exemplar Selection and Sensitivity Analysis

**Reference Pair Selection.** During training, reference input–output pairs $(x_r, y_r)$ are sampled uniformly at random from the dataset. We intentionally avoid curated or retrieval-based selection to mirror the intended test-time usage, where a user may provide *any* available exemplar without guarantees of optimal similarity or quality. At inference, GENIE operates under the same assumption and does not require task-specific or similarity-based reference selection.

**Sensitivity to Exemplar Quality and Semantic Mismatch.** GENIE is empirically robust to variations in exemplar domain, appearance, and semantic content, as illustrated in Fig 7. The model successfully transfers the intended transformation even when the reference and query images content differ significantly in visual style or domain (e.g., natural images as reference and medical or clipart images as query). GENIE does not require strict semantic alignment between reference and query content; instead, it relies on consistency of the *transformation* itself.

**Alternative K-shot Explicit Conditioning.** GENIE is inherently designed around a fixed $2 \times 2$ grid structure that encodes a single reference input–output pair and a single query. As a result, directly increasing the number of reference exemplars would require architectural changes and retraining. As an alternative that does not modify the grid structure, GENIE can incorporate multiple exemplars through explicit visual guidance. Given $K$ reference pairs, we compute individual transformation vectors $v_r^{(k)} = \varepsilon_\psi(y_r^{(k)}) - \varepsilon_\psi(x_r^{(k)})$ and aggregate them by averaging to obtain a single task representation $\bar{v}_r = \frac{1}{K} \sum_{k=1}^{K} v_r^{(k)}$. This aggregated vector is then used to form the explicit condition $c_v = \varepsilon_\psi(x_q) + \bar{v}_r$. As shown in Fig. 8, averaging explicit visual conditions across multiple reference pairs improves performance by approximately 1.5%, with the largest gains observed at small $K$.

| Out of Distribution (OOD) Task on OOD Data | | | | | | | |
|---|---|---|---|---|---|---|---|
| | Contrast Enhancement | | | Deraining | | | Map-to-Aerial | Scribble-to-Realistic |
| Models | PSNR ↑ | SSIM ↑ | LPIPS ↓ | PSNR ↑ | SSIM ↑ | LPIPS ↓ | LPIPS ↓ | LPIPS ↓ |
| Visual Prompt (Bar et al., 2022) | 12.090 | 0.308 | 0.929 | 14.983 | 0.391 | 0.473 | 0.823 | 0.614 |
| Painter (Wang et al., 2023b) | 4.127 | 0.004 | 0.871 | 6.024 | 0.007 | 0.743 | 0.863 | 0.797 |
| LVM (Bai et al., 2024) | 16.440 | 0.533 | 0.619 | **18.112** | 0.454 | **0.310** | 0.784 | 0.526 |
| GENIE | **18.513** | **0.671** | **0.585** | 17.414 | **0.513** | 0.357 | **0.721** | **0.517** |

Table 4: **Quantitative results of models on OOD Tasks**. Metrics marked with "↓" indicate lower is better, and metrics marked with "↑" indicate higher is better. The hyphen symbol "-" indicates that the model outputs unrelated or ambiguous results, making quantitative evaluation unsuitable. Best results are in "bold" and second-best are underlined.

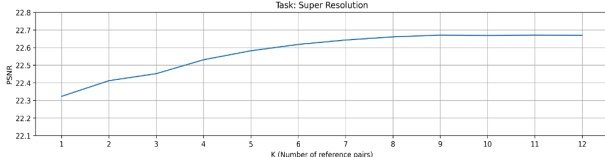

Figure 8: PSNR improves as explicit reference visual conditions are averaged across $K$ different reference pairs.

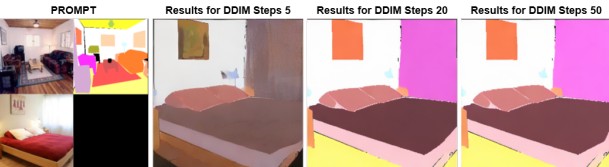

Figure 9: Segmentation result (color coded for each class) for different DDIM steps.

## 4.6  Inference Time

GENIE runs for ∼8s with 50 ddim steps, ∼3s with 20 ddim steps, and ∼1s with 5 ddim steps on a single A100 GPU. As seen in Fig. 9, 20 and 50 steps are qualitatively the same, so inference time is ∼3s per image. This performance is on par with any typical diffusion model.

## 4.7  Ablation Study

To isolate the contribution of GENIE's key components, we conducted an ablation study, summarized in Table 5. We evaluated four model variants on five representative tasks (three ID data on ID task and two OOD data on OOD data), starting from a minimal baseline to the full architecture.

Our study starts with **Model A**, which uses only implicit conditioning (ControlNet). We then introduce our proposed components individually: **Model B** adds the **task consistency loss**, which significantly boosts performance on generalization-heavy tasks like depth estimation and OOD transformations. **Model C** adds **explicit visual conditioning**, which excels on content-sensitive tasks such as inpainting and colorization. The full architecture, **Model D**, integrates both explicit conditioning and the task consistency loss. This complete model achieves superior results across all tasks, yielding an average performance improvement of **6.8%** over the next-best variant. This outcome confirms that explicit visual guidance and semantic consistency are complementary, validating GENIE's integrated design.

| Model | Explicit Conditioning | Task Consistency | ID Tasks on ID Data | | | OOD Task on OOD Data | |
|---|---|---|---|---|---|---|---|
| | | | Depth Estimation (RMSE ↓) | Colorization (MSE ↓) | Inpainting (SSIM ↑) | Contrast Enhancement (PSNR ↑) | Deraining (SSIM ↑) |
| **A** | × | × | 0.081 | 0.023 | 0.519 | 14.477 | 0.452 |
| **B** | × | ✓ | 0.069 | 0.021 | 0.597 | 16.789 | 0.504 |
| **C** | ✓ | × | 0.073 | 0.017 | 0.613 | 16.343 | 0.481 |
| **D** | ✓ | ✓ | 0.061 | 0.016 | 0.632 | 18.513 | 0.513 |

Table 5: **Ablation study results across selected tasks**. Models: A) Implicit conditioning only, B) Implicit conditioning + task consistency, C) Implicit + explicit conditioning, D) GENIE.

## 5  Conclusions & Future Directions

We introduced GENIE, a unified diffusion framework for text-free multi-task visual prompt learning. Driven by its dual visual conditioning and a novel task consistency loss, GENIE achieves generalization and fidelity across diverse ID and OOD data and tasks, outperforming state-of-the-art visual-only prompting baselines.

Despite its strong performance, GENIE shows limitations, particularly in segmentation where latent-space modeling can cause label misalignment, and on highly divergent OOD tasks (Fig. 10). Future work could address this by incorporating pixel-level supervision for generated output and expanding the training data to further enhance GENIE's adaptability for real-webm vision challenges.

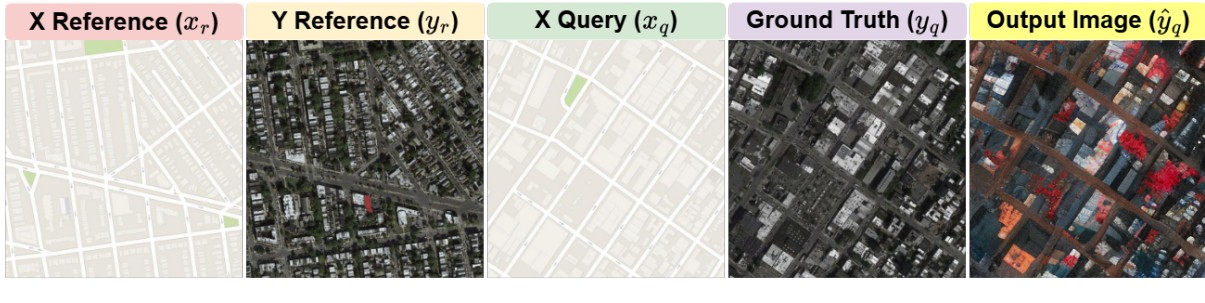

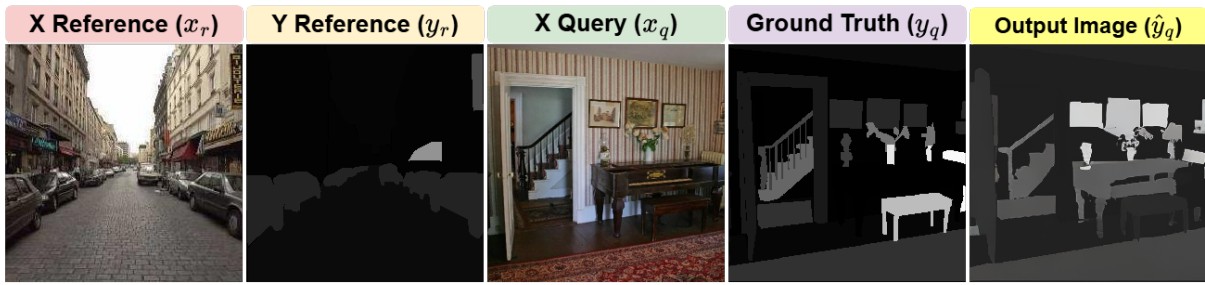

Figure 10: **Qualitative results showing GENIE's limitations. Top:** An OOD task on OOD data, transitioning from map-style to aerial satellite imagery from (Isola et al., 2017). GENIE attempts to fill the map and results in ambiguous and unrealistic outputs. **Bottom:** A segmentation task where GENIE struggles with class label assignments, producing inconsistent segmentation masks.

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

# Appendix

## A    Introduction

This supplementary material provides additional analysis of GENIE's explicit conditioning mechanism, its performance on out-of-distribution (OOD) datasets and tasks, and an examination of its limitations. In Section B, we detail the design choices behind explicit conditioning, including our rationale for selecting CLIP-ViT Radford et al. (2021) as the visual encoder over alternatives such as VLMs Liu et al. (2024a), VQ-VAE Esser et al. (2021), and ViT Dosovitskiy et al. (2021). Section C outlines the datasets used for training and evaluation, highlighting their characteristics and relevance to task diversity. Section D presents qualitative results on OOD datasets from varied visual domains, including Clipart, Painting, and Sketch subsets of DomainNet, showcasing GENIE's generalization ability. This section also includes failure cases, focusing on GENIE's challenges in extreme OOD tasks and semantic segmentation scenarios.

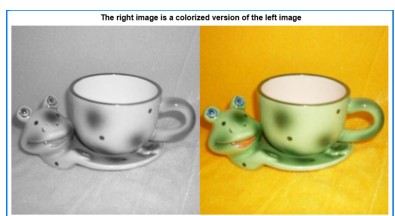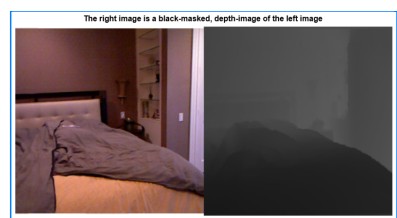

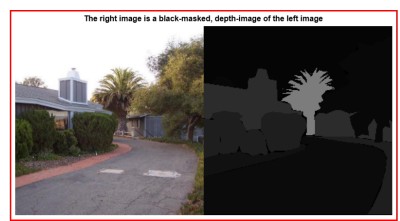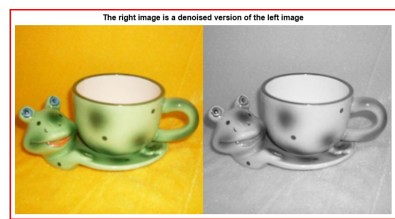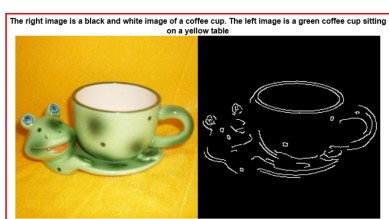

Figure 11: The figure illustrates the result of LLaVA Liu et al. (2024a) (VLM) in generating captions for reference pair transformations. Blue-bordered images (top) represent correctly generated captions. Red-bordered images (bottom) indicate incorrect captions, where the VLM misinterprets the transformation.

## B    Explicit Conditioning Choice

### B.1    Vision-Language Models (VLMs) for Task Transformation Understanding

We explored VLMs, specifically LLaVA Liu et al. (2024a), to We explored the feasibility of interpreting transformations between reference image pairs $(x_r, y_r)$ in a zero-shot setting using a vision-language model (VLM). The goal was to replace predefined, task-specific text prompts with captions generated by the VLM that describe the transformation from input to output in the reference pair. These captions would then be passed through the CLIP text encoder to provide conditioning information within the stable diffusion pipeline.

To facilitate this, the VLM was prompted with the following instruction:

***Caption:*** *The image is a concatenation of two images side by side. Tell me the relationship between the images. You can instruct like "The right image is* `<task>` *of the left image." Choose the closest* `<task>` *from: "Segmentation," "Denoised," "Colorization," "Hed map," "Boundary image," "Grayscale," "Black-Masked," "Depth-image" based on your knowledge.*

The VLM performed reasonably well on simpler transformations. For instance, it correctly identified colorization tasks with captions like "The right image is a colorized version of the left image." However, it frequently failed to recognize more complex transformations such as segmentation, edge detection, and masked-image reconstruction (Fig. 11). In these cases, the model often produced ambiguous or incorrect descriptions—e.g.,

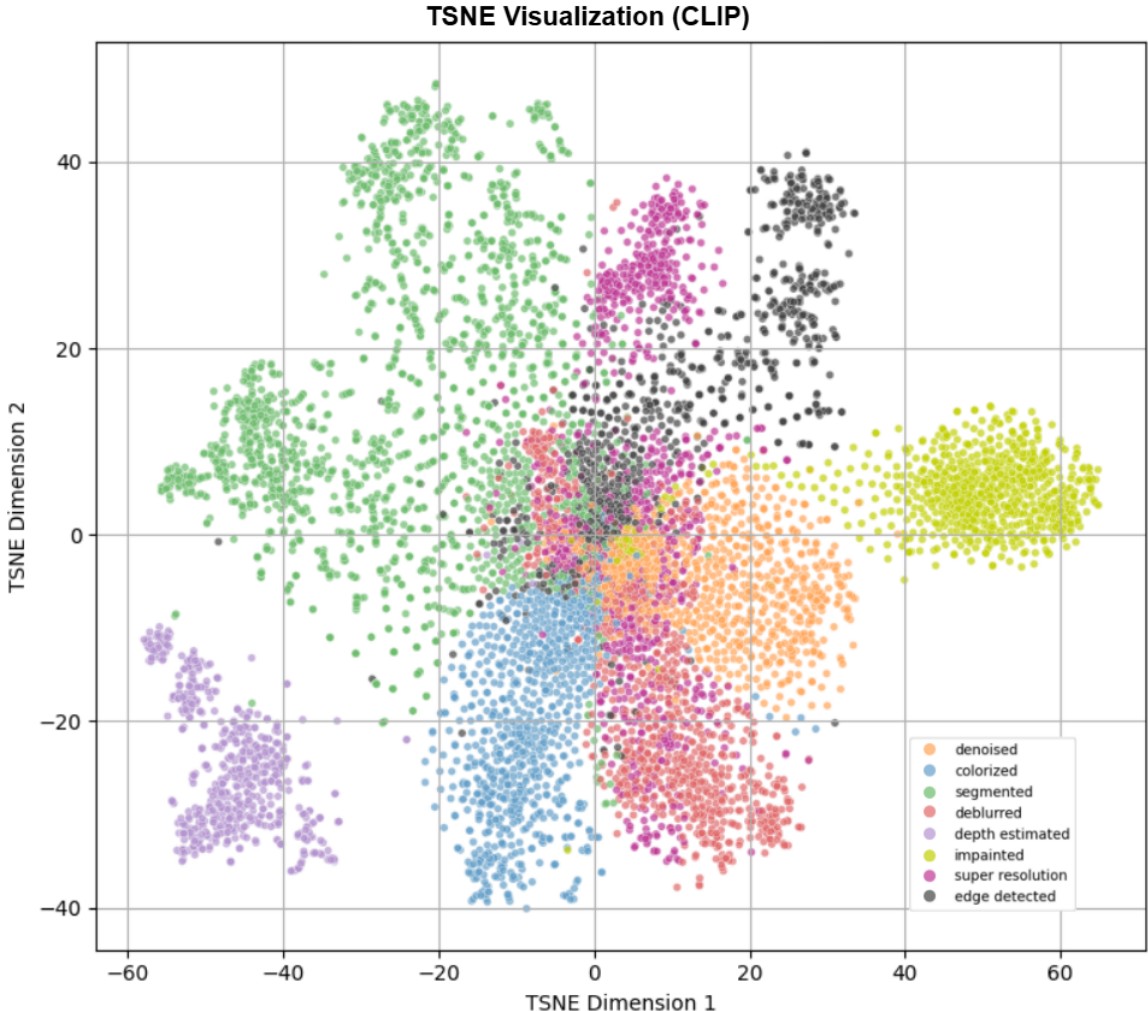

Figure 12: **t-SNE Visualization of CLIP-ViT task-specific embeddings.** The embeddings represent differences between the transformed image and the original image for each task. CLIP-ViT provides well-separated task clusters.

interpreting edge detection as "black and white image" or generating generic captions lacking task-specific clarity.

These limitations stem from the VLM's training objectives, which prioritize general content understanding over fine-grained task-specific transformations. While effective for high-level semantic alignment, this makes VLMs unreliable for providing precise task context. As a result, we conclude that such captioning mechanisms are inadequate for replacing textual prompts in the CLIP text encoder within our framework. Consequently, we shift our focus to visual encoders for explicit conditioning, aiming to fully eliminate textual dependencies in GENIE.

## B.2   Encoder Evaluation

To identify the most suitable visual encoder for capturing task-specific transformations in our framework, we experimented with three encoders: CLIP-ViT Radford et al. (2021), ViT Dosovitskiy et al. (2021), and VQ-VAE. Each encoder's embeddings were analyzed using t-SNE van der Maaten & Hinton (2008) to visualize how well the embeddings differentiate various transformation tasks such as deblurring, masking,

noise removal, and edge detection. The t-SNE projections represent the **difference embeddings**, which capture the difference between the transformed image and the original image for each task.

### B.2.1 Task-Specific Embeddings

- **CLIP-ViT and ViT**: We evaluated both CLIP-ViT and ViT in zero-shot settings, and their respective t-SNE projections (Figures 12 and 13) reveal well-separated task clusters, indicating their effectiveness in capturing task-specific visual semantics. While their performance was comparable, we chose CLIP-ViT for our framework. This decision is motivated by the fact that CLIP's visual and textual encoders are jointly trained using contrastive learning, which aligns the visual embedding space with text semantics. As a result, using CLIP-ViT to replace the textual encoder in the diffusion model's conditioning pathway ensures greater compatibility and semantic consistency within the latent diffusion architecture.

- **VAE**: In contrast, the t-SNE projection for VAE-based embeddings (Fig. 14) shows substantial overlap between clusters, particularly for visually similar tasks such as deblurring and super-resolution. This poor task separability suggests that VAE embeddings lack the discriminative capacity to model fine-grained task transformations, making them unsuitable for use in GENIE's conditioning pipeline.

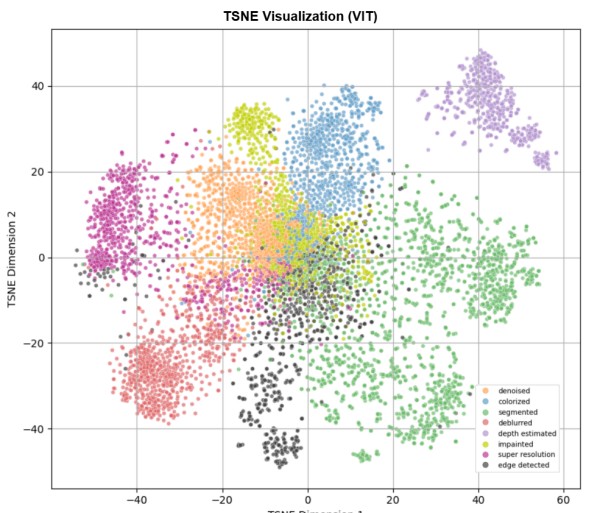

Figure 13: **t-SNE Visualization of ViT task-specific embeddings.** Task clusters are well-separated, demonstrating strong feature extraction capabilities.

Figure 14: **t-SNE Visualization of VAE task-specific embeddings.** The embeddings show significant overlap between tasks, particularly deblurring and super-resolution.

Although the learned task-specific embeddings capture meaningful task semantics, they may also contain irrelevant information such as content-related features or noise in a few of the tasks. We rely on training to help the model suppress this noise and emphasize task-relevant signals. As part of future work, we plan to improve the robustness of these embeddings by explicitly disentangling task-specific information from content and noise.

### B.2.2 Content-Specific Embeddings

We further evaluated whether Diffusion-VAE and CLIP-ViT can effectively cluster images based on object information, with each cluster corresponding to a distinct object class. If an encoder can successfully cluster objects, it indicates that the encoder is capable of deciphering the semantic information about the object. This ability is crucial for conditioning the diffusion model in downstream tasks, as the extracted object semantics and transformations can be used to guide the generation process.

| Dataset | Number of Training Samples | Number of Testing Samples | Tasks |
|---|---|---|---|
| **ImageNet** | 20000 per task | 20000 | Impainting, Deblurring, Colorization, Edge Detection, Super-Resolution, Denoising |
| **ADE 20K** | 20000 | 3000 | Segmentation |
| **NYU V2** | 20000 | 654 | Depth Estimation |
| **LOL** | - | 15 | Contrast Enhancement |
| **Deraining** | - | 500 | Deraining |

Table 6: Summary of datasets used for training and evaluating GENIE, along with their associated tasks. The " - " symbol indicates datasets that were exclusively used for evaluation and not included in GENIE's training pipeline.

As seen in Fig.15, Diffusion-VAE embeddings show significant overlap between object classes, indicating a struggle in capturing object-specific features. This limitation suggests that Diffusion-VAE may not be well-suited for conditioning the diffusion model based on object semantics. In contrast, Fig.16 demonstrates that CLIP-ViT embeddings form distinct clusters for each object class. This clustering suggests that CLIP-ViT is capable of extracting the necessary features and can be further used to condition a diffusion model.

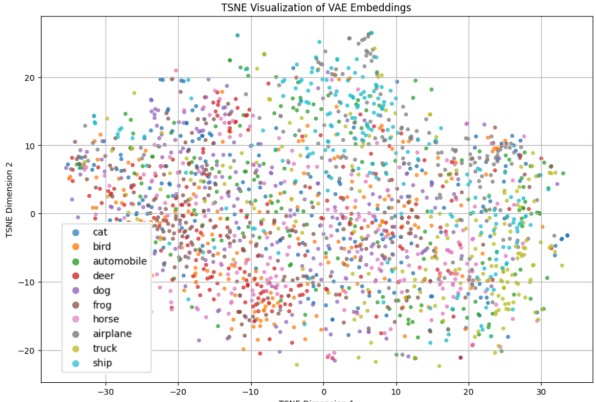

Figure 15: **t-SNE Visualization of VAE content-specific Embeddings.** The embeddings show significant overlap between object classes, indicating limitations in capturing object-specific features.

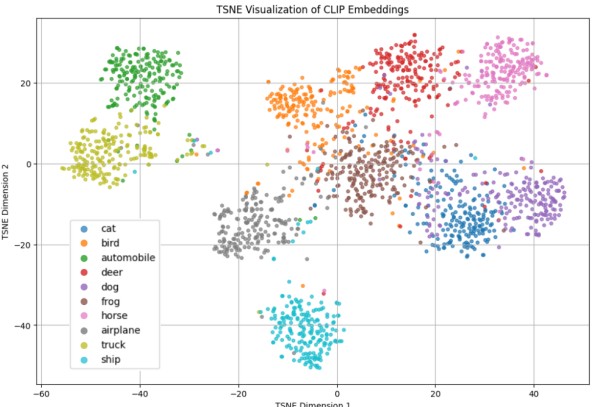

Figure 16: **t-SNE Visualization of CLIP-ViT content-specific Embeddings.** The embeddings form distinct clusters for each object class, indicating effective extraction of object semantics.

## C Datasets

### C.1 In-Distribution Training Dataset

**ImageNet-22K:** ImageNet-22K Deng et al. (2009) is a large-scale, hierarchical visual dataset containing over 14 million images categorized into 22,000 classes.

**ADE20K:** ADE20K Zhou et al. (2017) is a scene-centric dataset comprising over 25,000 images annotated for 150 semantic categories, including both indoor and outdoor environments. It is widely used for segmentation tasks, with 20,000 images allocated for training, 2,000 for validation, and 3,000 for testing .

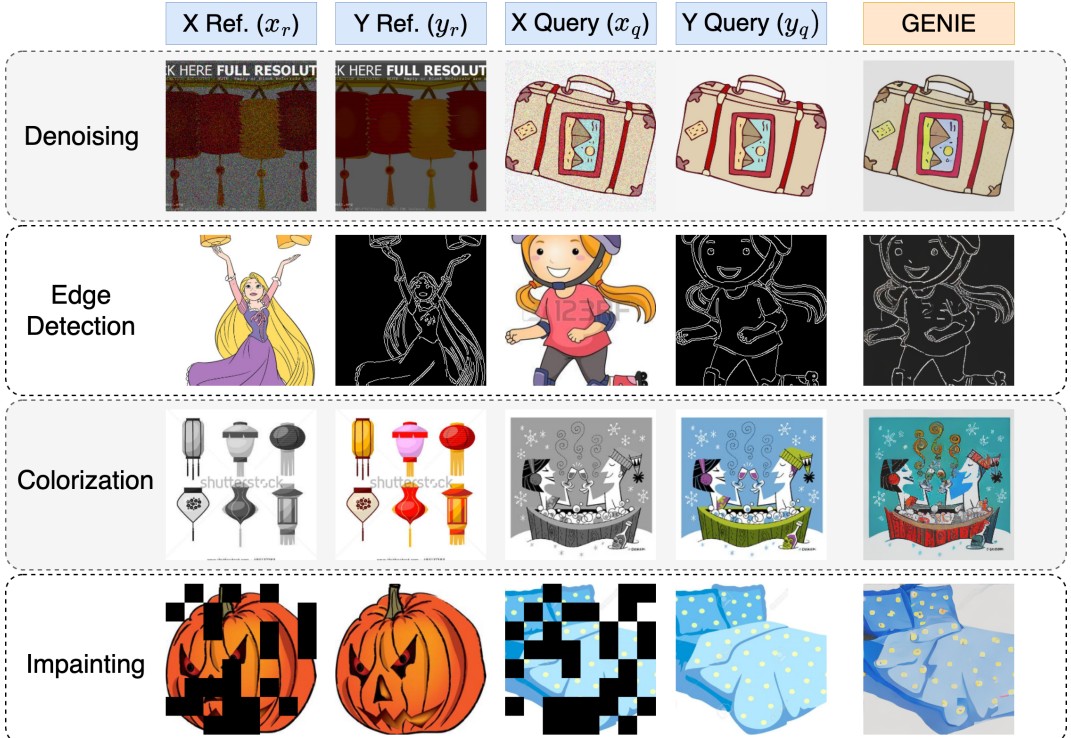

Figure 17: **Qualitative result of GENIE on various tasks on CLIPART domain from domain net Peng et al. (2019) dataset.**

**NYU Depth V2:** NYU Silberman et al. (2012) Depth V2 contains RGB-D images captured in 464 indoor scenes using a Microsoft Kinect camera. It provides dense depth annotations alongside RGB data, with 24,000 images in the training set and 654 images in the test set.

### C.2 Out-of-distribution (OOD) Testing Dataset

**LOL:** The Low-Light Dataset (LOL) Wei et al. (2018) consists of paired low-light and enhanced images, allowing models to learn to enhance visibility and detail in poorly lit scenes. This dataset is commonly used for low-light image enhancement tasks. It contsist a total of 500 image pairs, split into 485 images for training and 15 pairs for testing.

**DomainNet:** DomainNet Peng et al. (2019) is a large-scale dataset spanning six diverse domains, including real, sketch, clipart, and painting. Since our model is trained on ImageNet Deng et al. (2009), NYUv2 Silberman et al. (2012), and ADE20k Zhou et al. (2017), we perform qualitative evaluations on the painting, sketch, and clipart domains. These domains differ significantly from the natural image datasets used during training, offering an opportunity to assess our model's adaptability.

**Deraining Dataset:** The Deraining dataset Zamir et al. (2022) contains images with rain streaks and their corresponding clear versions. It is used to test how well a model can remove the rain and restore clear images. This dataset is a common benchmark for evaluating performance in OOD scenarios. The dataset consists of 500 images for training and 500 images for testing. In our experiments, we utilized all 500 test images to test our model.

## D Qualitative Results

This section provides qualitative examples of GENIE's performance across a range out-of-distribution (OOD) data. figures 17, 19, and 18 display results on three different domains from DomainNet Peng et al. (2019). These examples highlight the model's capacity to generalize task transformations under OOD data scenarios.

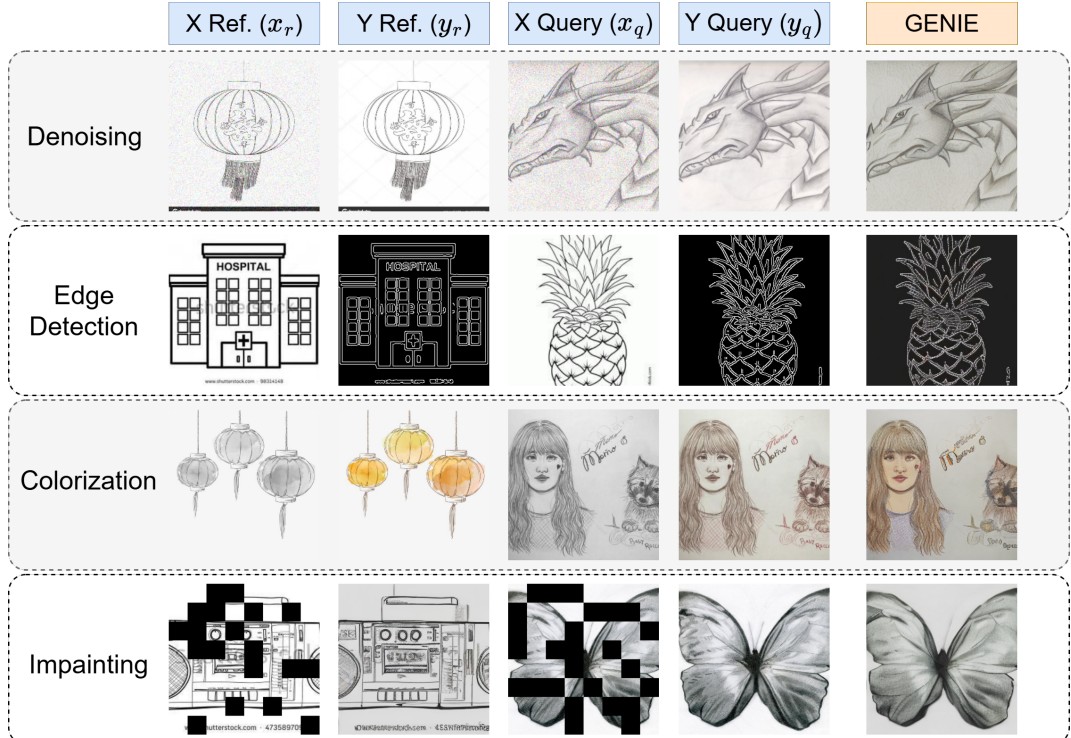

Figure 18: Qualitative result of on various tasks on SKETCH domain from domain net Peng et al. (2019) dataset.

While GENIE demonstrates robust performance in some OOD tasks (as shown in Fig.20), Fig.21 highlights some of the model's limitations. These includes challenges in accurately segmenting objects into the correct classes as intended in the ADE20k dataset, challenges in deciphering scribble inputs to generate realistic outputs, and the inability to capture intricate details of pose estimation tasks from the reference image. These examples showcase areas where further refinement is needed, particularly in handling complex input-output transformations and finer task intricacies.

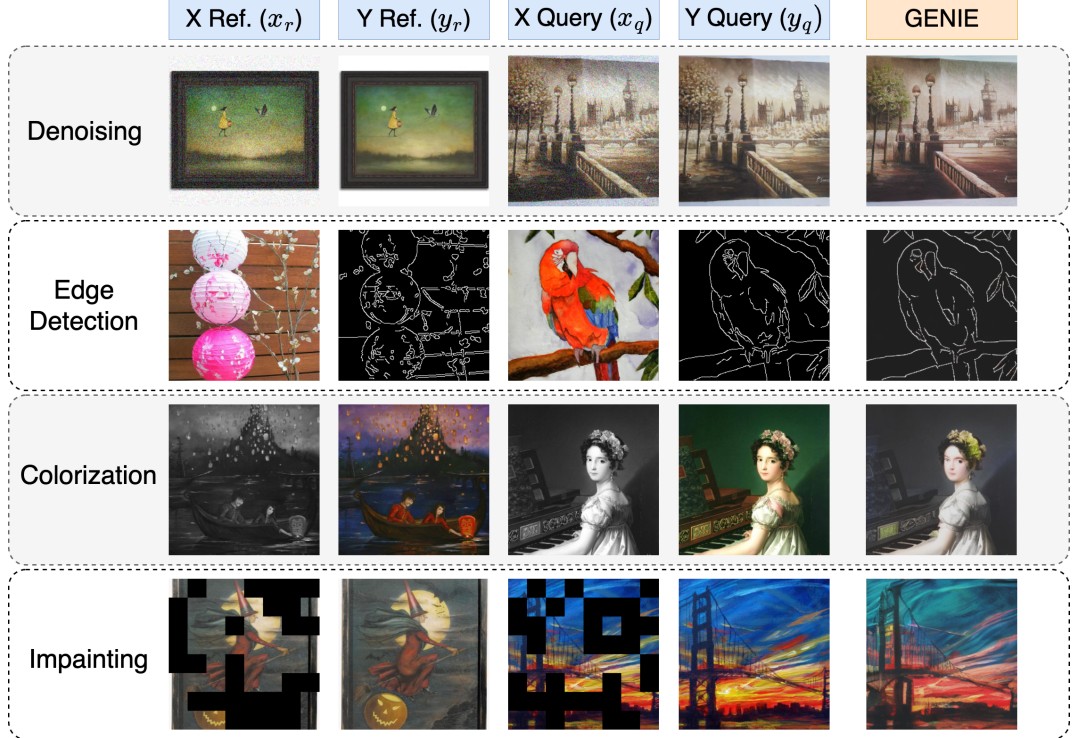

Figure 19: Qualitative Result result of GENIE on various tasks on PAINTING domain from domain net Peng et al. (2019) dataset.

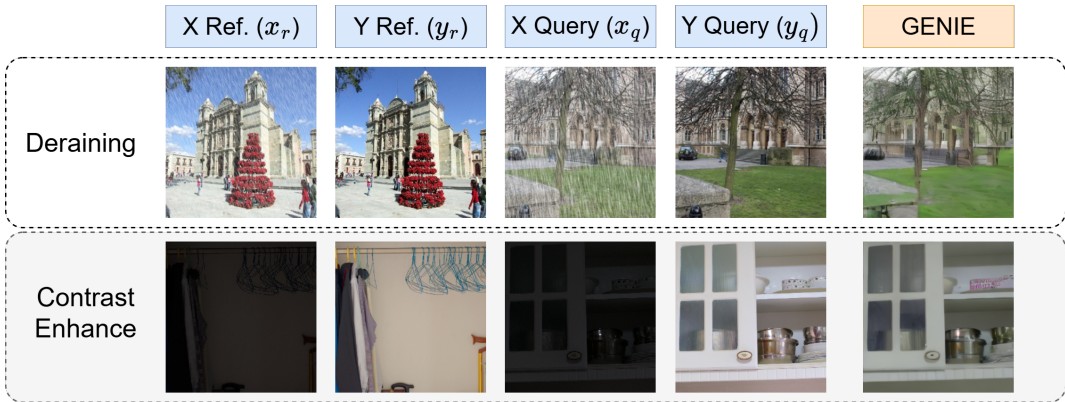

Figure 20: Qualitative result on Out-of-distribution tasks.

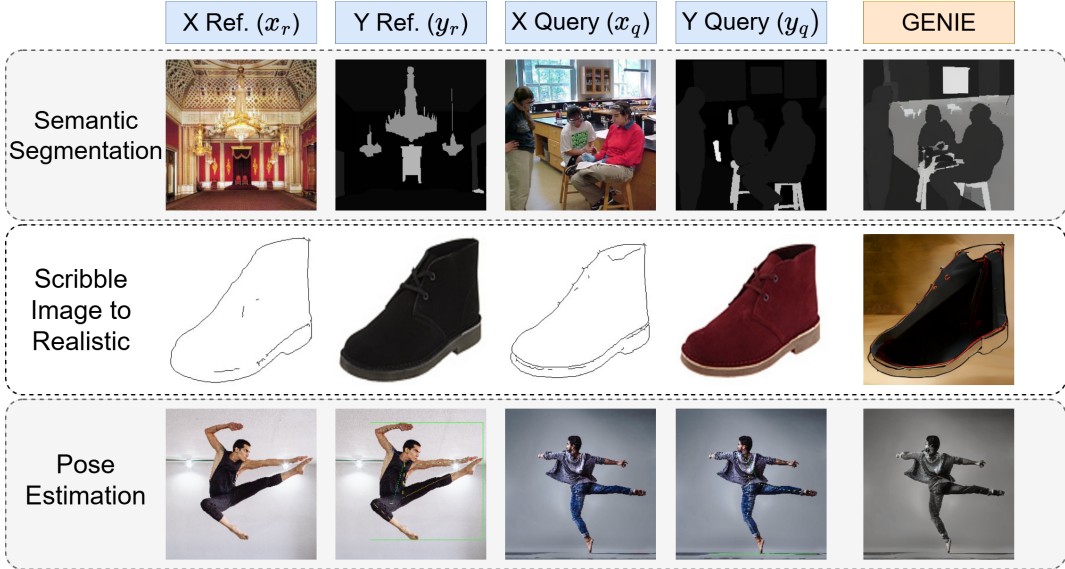

Figure 21: **Limitations of GENIE:** The figure illustrates limitations in semantic segmentation where though the model is able to segment the objects in scene but it struggles to assign the class as intended in the dataset, challenges in generating realistic shoes from scribble input by filling with random colors, and difficulties in capturing fine details for pose estimation from the reference image.

