# OpenReview forum: "GENIE: A Visual-Only Diffusion Framework for Task- Agnostic Image Transformation"
_TMLR — Accepted by TMLR_

### Review · Reviewer_hUi6 · 2026-01-02

**Summary Of Contributions:**

This paper introduces GENIE, a framework for task-agnostic image transformation using visual-only diffusion models. GENIE extends ControlNet by operating on a 2 x 2 image grid consisting of a reference input, reference output, query input, and a masked query output. The model predicts the query output by inferring the transformation implied by the reference input-output pair, enabling tasks such as colorization, denoising, and enhancement without textual supervision.

The approach combines implicit visual conditioning through ControlNet with explicit task conditioning derived from CLIP embedding differences between the reference input and output. During training, a task consistency loss encourages alignment between the CLIP-space transformation applied to the reference pair and that applied to the query pair. The paper compares GENIE against three visual-only baselines and reports competitive performance on both in-distribution and selected out-of-distribution (OOD) tasks.

**Audience:**

Yes

**Audience Explanation:**

GENIE presents a simple and intuitive extension of ControlNet for visual-only, task-agnostic image transformation. The use of explicit task conditioning via CLIP features and a task consistency loss is well-motivated and supported by ablation studies. The method’s simplicity, combined with its empirical performance on in-distribution tasks and exploratory results on OOD settings, should be of interest to readers working on generative image modeling, diffusion models, and generalization across visual tasks.

**Claims And Evidence:**

No

**Claims Explanation:**

- The paper provides clear and convincing evidence for in-distribution performance, including comparisons against multiple baselines and a well-structured ablation study that justifies key design choices.
- However, the strength of the evidence weakens for the paper’s OOD generalization claims:
  - The contrast enhancement OOD task, where GENIE achieves its strongest reported gains, is evaluated on only 15 test images, which is insufficient to draw robust conclusions about generalization.
  - For the deraining OOD task, GENIE performs similarly to LVM, and it is unclear whether the proposed method provides a consistent or meaningful improvement.
  - Painter (Wang et al., 2023b) is not reported for OOD tasks in Table 4. Although Painter demonstrates OOD generalization when provided with explicit visual task prompts for known tasks, it is not designed to infer task definitions from exemplar input-output pairs as required in this setting. As a result, its outputs are often semantically invalid under the exemplar-based OOD evaluation considered here. A clearer discussion of this mismatch, or reporting Painter’s results with appropriate caveats, would improve the clarity and interpretability of the comparison.
  - The paper presents several exemplar-based transformations (e.g., scribble generation and map-to-aerial translation) in Figures 2, 3, and 8, as well as Appendix Figures 15-19, as evidence of task-agnostic and OOD capabilities or limitations. However, these results are shown only qualitatively, without quantitative metrics and without baseline comparisons for Figure 8 and the Appendix figures. While visually compelling, these examples are anecdotal and do not provide convincing evidence of systematic OOD generalization or failure.

**Requested Changes:**

- The paper makes claims about OOD generalization but reports quantitative results for only two OOD tasks, compared to seven in-distribution tasks. Adding more OOD tasks with quantitative metrics, larger test sets, or variance estimates would substantially strengthen these claims.
- The paper matches only the magnitude of CLIP-space transformation vectors. It would be helpful to explain why magnitude matching was chosen over alternatives such as directional alignment (e.g., cosine similarity) or direct vector matching (e.g., L1/L2 losses), and whether such alternatives were explored.
- If GENIE does not generate valid class-labeled outputs for segmentation, it is unclear why metrics are reported in this setting in Table 3. An alternative evaluation that does not penalize class mislabeling (eg. matching predicted regions to ground truth based on IoU) could provide a more meaningful assessment and alleviate concerns about fairness and interpretability.
- Add relevant discussion about why quantitative results are not reported for OOD tasks for Painter in Table 4 to provide context for readers to interpret results.

---

> ### Author Response · Authors · 2026-01-20
> **Segmentation metric clarification and evaluation**
>
> We thank the reviewer for raising this important point regarding the fairness and interpretability of segmentation metrics. As discussed in Section 4.4(a), GENIE is able to correctly segment and group multiple objects across complex scenes and domains (Figs. 5, 6, and 7 of the original paper) but occasionally assigns incorrect class labels due to the absence of explicit pixel-level supervision. This behavior is also observed in latent-space and visual-prompting models (e.g., LVM and Visual Prompting) and reflects an optimization toward semantic structure rather than exact pixel-wise labeling, making metrics such as mIoU an incomplete indicator of segmentation quality in this setting.
>
> To better align the evaluation with the model’s capabilities and address the reviewer’s concern, we will report the Adjusted Rand Index (ARI) in Table 3. ARI is a clustering-based, label-agnostic metric that measures the consistency between predicted regions and ground-truth segmentations by evaluating whether pixels belonging to the same object are grouped together, without penalizing incorrect class assignments. This provides a more meaningful assessment of structural segmentation quality when semantic labeling is imperfect.
>
> After computing the Adjusted Rand Index (ARI) for segmentation, we observe that GENIE achieves the second-best performance among the compared methods, trailing Painter by approximately 5%. We include this metric in the revised table for completeness and ease of comparison. In addition, we clarify in the revised manuscript that GENIE achieves an 8.30% improvement in performance on in-distribution tasks and data.
>
> Final performance metric is as follow:
> | Metric (higher is better) | Visual Prompt | Painter | LVM  | GENIE |
> |--------------------------|---------------|---------|------|-------|
> | ARI                      | 0.303         | 0.650   | 0.573| 0.613 |

---

> ### Author Response · Authors · 2026-01-20
> **Loss Function Clarification**
>
> We thank the reviewer for highlighting this point and apologize for the confusion caused by our earlier description which was a typographical error from our end in the original paper. To clarify, our method does not match only the magnitude of CLIP-space transformation vectors. Instead, we compute the difference between the CLIP embeddings of the reference input–output pair ($x_r, y_r$) to obtain a task-specific displacement vector in CLIP space. We then enforce the same task shift by directly matching this displacement to the difference between ($x_q, \hat{y}_q$) using an $L_2$ loss, thereby performing full vector matching. Final Formulation is as follow:
>
> $v_r = \varepsilon_\psi(y_r) - \varepsilon_\psi(x_r)$ and $v_q = \varepsilon_\psi(\hat{y_q}) - \varepsilon_\psi(x_q) $ and finally
>
> $$
> \mathcal{L}_T = \mathbb{E} \left[ \left\|| v_r - v_q \right\||_2^2 \right]
> $$
>
> We have revised the manuscript to explicitly state this formulation.

---

> ### Author Response · Authors · 2026-01-20
> **Clarification of painter for OOD task on OOD data performance**
>
> As illustrated in Fig. 5 of the original paper, Painter struggles to produce semantically valid outputs for exemplar-based OOD tasks on OOD data, as it is unable to infer task information from input–output pairs that it has not seen during the training. Consequently, its predictions often do not correspond to valid task executions, making quantitative metrics misleading. For this reason, we omit Painter from the OOD quantitative comparison and will add a dedicated discussion explicitly stating this limitation in Section 4.4 (b) to improve clarity and interpretability for readers. For your reference, we are adding the performance metric of painter on the original OOD task on OOD data:
> | Task | Contrast Enhancement |        |        | Deraining |        |        |
> |---------|----------------------|--------|--------|-----------|--------|--------|
> |         | PSNR                 | SSIM   | LPIPS  | PSNR      | SSIM   | LPIPS  |
> | Painter | 4.127                | 0.0044 | 0.871  | 6.024     | 0.067  | 0.7426 |

---

> ### Author Response · Authors · 2026-01-20
> **Additional OOD task evaluation**
>
> We thank the reviewer for this comment and conducted additional quantitative evaluations for the exemplar-based map-to-aerial translation and scribble generation tasks that were previously shown only qualitatively. Specifically, we evaluate map-to-aerial translation on 1,000 images and scribble generation on 2,000 images, and report LPIPS scores for all compared models.
>
> | Method  | Map to Aerial |    Scribble Generation    |
> |---------|----------------------|--------|
> |         | LPIPS | LPIPS |
> | Visual Prompt | 0.823 | 0.614 |
> | Painter | 0.863 | 0.797 |
> | LVM | 0.784 | 0.526 |
> | GENIE | 0.721| 0.517 |
>
> The results show that GENIE consistently outperforms LVM, achieving approximately an 8% improvement on map-to-aerial translation and a 1.7% improvement on scribble generation. Overall, we achieve an improvement of 5.96% compared to the next best performing model in OOD task on OOD data. These quantitative results are now included in the revised manuscript to complement the qualitative figures and to provide more systematic evidence of GENIE’s exemplar-conditioned generalization behavior.

---

### Review · Reviewer_KoGj · 2026-01-02

**Summary Of Contributions:**

The paper introduces GENIE, a unified ControlNet–diffusion framework for task-based image generation driven purely by visual exemplars (reference input–output pairs), removing the need for text prompts or auxiliary metadata.
GENIE infers task intent via a dual visual conditioning scheme: implicit guidance through ControlNet and explicit task encoding using CLIP-based visual arithmetic to capture the transformation implied by the exemplar pair.
To better align semantics between exemplars and generated results, it proposes a lightweight task consistency loss that enforces embedding-space coherence across transformed pairs.
Without any task-specific architectural changes or loss redesign, GENIE supports task switching across many transformations and shows strong scalability, achieving about 10% average improvement over visual-conditioned baselines across seven standard tasks and generalizing to two OOD tasks (deraining and contrast enhancement).

**Audience:**

Yes

**Audience Explanation:**

The task of visual-conditional image processing is a new and vital area, which I believe will be interesting to a wide audience.

**Claims And Evidence:**

Yes

**Claims Explanation:**

This paper has included sufficient experiments showing the qualitative and quantitative comparisons against the baselines. The method details are also well illustrated.

**Requested Changes:**

### Weaknesses / Questions

#### Minor issues

- There is a duplicated phrase (“that the that the”) that should be corrected.

- The manuscript uses “\cite”. Please replace with a more proper citation such as "\citep".

#### Technical questions and concerns

- The paper appears to use a pretrained VQGAN for encoding/decoding image grids formed by concatenating multiple images.
Since VQGAN is typically pretrained on natural single images rather than tiled grids, could this distribution shift degrade reconstruction fidelity or downstream generation quality?
It would be helpful to (a) justify this design choice, or (b) provide a small quantitative check (e.g., reconstruction error or downstream performance with/without grid encoding, or with a VQGAN trained/fine-tuned on grids).

- The task consistency loss is described as a pixel-level difference between embeddings, which is confusing. Please clarify the exact formulation and provide intuition for why this objective is appropriate for enforcing task consistency. Also, the use of 50% dropout as a mitigation seems largely empirical. Could the authors provide more principled alternatives or justification?

- Geometric claim needs stronger support: The statement "Geometrically, it enforces that vr and vq lie on the same hypersphere in E, preserving the geodesic radius
of the transformation. Operating in a contrastively structured feature space, this avoids both under- and
over-editing, thereby supporting generalization in both ID and OOD settings where task style may vary but
transformation strength should remain coherent."

- Since the method relies on reference input–output pairs, performance may depend on how these exemplar pairs are selected. Please clarify: a) What criteria are used to select reference pairs at test time? b) How sensitive is GENIE to exemplar quality, semantic mismatch, or transformation strength? c) Is there a recommended retrieval strategy and does retrieval introduce extra overhead? d) An ablation on exemplar selection (random vs. curated vs. retrieval-based) would make the practical usage clearer.

- Competitiveness vs SOTA: While the framework is interesting and flexible, the reported results do not consistently establish state-of-the-art performance on standard restoration tasks. The paper would benefit from clearer positioning: is the main goal unified task switching and text-free control, rather than outperforming specialized methods?

I acknowledge the novelty of the work and after the aforementioned concerns are addressed, I am lean to acceptance.

---

> ### Author Response · Authors · 2026-01-20
> **VQGAN grid formulation clarification**
>
> We thank the reviewer for raising this important concern regarding potential distribution shift when applying a pretrained VQ-GAN to concatenated image grids. Although VQ-GAN is typically pretrained on natural single images, we use it to encode and decode concatenated 2×2 image grids to enable unified processing of visual examples. This design was similar to the tokenization process of (Visual prompting via image inpainting Bar et al. 2022). To verify that this design choice does not introduce significant reconstruction artifacts, we conducted a quantitative evaluation of reconstruction fidelity on grid inputs using the pretrained VQ-GAN without fine-tuning. Across 10,000 samples, we observe a mean L1 reconstruction error of **0.033** and a mean RMSE of **0.073**, indicating that the grid structure and visual content are well preserved.
> | **$L_1$ Error** | **$L_2$ Error** |
> |-------|-------|
> | 0.033 | 0.073 |
>
> For your reference, we will add the qualitative result for the same in the updated paper that demonstrates preservation of spatial layout and image details. Together, these results suggest that the pretrained VQ-GAN generalizes sufficiently to tiled grid inputs and does not adversely impact downstream generation quality.

---

> ### Author Response · Authors · 2026-01-20
> **Loss function clarification**
>
> We would like to clarify that the proposed task consistency loss is not a pixel-level objective. Instead, it is entirely defined in CLIP semantic embedding space. This confusion might possibly be caused by the typographical error which we will correct in the updated paper.
>
> $$
> \mathcal{L}_T = \mathbb{E} \left[ \left\|| v_r - v_q \right\||_2^2 \right]
> $$
>
> where $v_r = \varepsilon_\psi(y_r) - \varepsilon_\psi(x_r)$, $v_q = \varepsilon_\psi(\hat{y_q}) - \varepsilon_\psi(x_q)$
>
> Regarding conditioning dropout, we clarify that randomly dropping the explicit semantic condition $c_v$ during training follows established practice in prior diffusion models. Conditioning dropout is used in ControlNet (Zhang et al 2023) to reduce over-reliance on prompts and encourage semantic inference from visual inputs, and similar stochastic conditioning underlies classifier-free guidance in GLIDE (Ho & Salimans, 2022b). We have updated the paper to explicitly state this motivation.

---

> ### Author Response · Authors · 2026-01-20
> **Reliance on reference pair**
>
> **Reference selection at test time:** GENIE does not assume curated or retrieval-based exemplar selection. During training, reference input–output pairs are sampled uniformly at random, and the same assumption is made at inference, reflecting realistic usage where users may provide arbitrary exemplars without guarantees of optimal similarity.
>
> **Sensitivity to exemplar quality and semantic mismatch:** As shown in Fig. 6 of the original paper, GENIE is empirically robust to significant domain and appearance mismatch between reference and query images (e.g., natural images as reference and medical or clipart images as query). The model relies on consistency of the inferred transformation rather than semantic alignment of image content, enabling generalization across different domains.
>
> **Retrieval strategy and overhead:** GENIE does not employ any retrieval mechanism and introduces no additional inference overhead. The current framework assumes arbitrary exemplar availability at test time. We thank the reviewer for the suggestion of retrieval-based exemplar selection and as noted in Section 3.3 (a) of the original paper that this could be a complementary extension but is beyond the scope of the present work.
>
> In the updated paper, we will add a dedicated section to highlight this.

---

> ### Author Response · Authors · 2026-01-20
> **Competitiveness vs SOTA**
>
> Our primary contribution is a unified framework that enables task inference, switching, and execution using only visual paired prompts, without text supervision or other auxiliary cues. Accordingly, our state-of-the-art claims are made with respect to prior methods operating under the same setting—i.e., models that rely solely on visual input–output exemplars to infer and perform tasks. We would like to clarify that our state-of-the-art (SOTA) is not with respect to the models that specialise in the task that we have considered. We have revised the paper to more explicitly position our results within this context and to avoid any ambiguity regarding comparisons to task-specific or text-conditioned approaches.

---

> ### Author Response · Authors · 2026-01-20
> **Minor issue**
>
> We thank the reviewer for pointing out our mistake and improvement suggestion. We have made the necessary changes in the updated paper.

---

> ### Author Response · Authors · 2026-01-20
> **Geometric claim clarification**
>
> By inspecting the t-SNE visualization in Fig. 10 (Supplementary material of original paper), we observe that CLIP embeddings form well-separated, compact clusters by semantic category. While t-SNE is primarily illustrative, this structure provides useful intuition that semantic transformations correspond to coherent displacements in embedding space rather than arbitrary pixel-level changes. Motivated by this, we inject an **inductive bias** int the existing diffusion loss that models a task as the embedding-space displacement derived from the reference input–output pair and enforces a comparable displacement for the query. Operating on transformation vectors rather than pixel outputs allows GENIE to exploit the geometry of CLIP embedding space and utilizing the zero-shot capabilities of CLIP architecture.

---

### Review · Reviewer_pDuY · 2026-01-06

**Summary Of Contributions:**

**Summary:**

The paper proposes GENIE, a visual-only latent diffusion framework that performs image-to-image transformations by conditioning on a single reference input–output pair $(x_r, y_r)$ plus a query input $x_q$, arranged in a $2\times2$ grid with the target quadrant masked. GENIE combines implicit conditioning via ControlNet features from the grid, explicit conditioning via CLIP embedding arithmetic, and adds a task-consistency regularizer. The paper reports around 10% average gains over retrained vision-only baselines across 7 in-distribution tasks and 2 “OOD” tasks (deraining and contrast enhancement).

---

**Strengths:**

**1. Simple modeling recipe.** Exemplar grid conditioning + ControlNet + explicit CLIP “task vector” arithmetic is straightforward to implement and ablate.

**2. Broad task coverage.** The evaluation spans standard i2i tasks (e.g., inpainting, denoising, depth, segmentation) and includes additional robustness checks.

**3. Clear training scale and evaluation structure.** The paper states a consistent training scale (20k samples per task for the main training tasks) and organizes evaluation into ID tasks on ID data, OOD tasks from unseen datasets, and ID tasks on OOD domains.

**4. Acknowledged failure modes.** The paper discusses some limitations, such as segmentation label issues and qualitative failures under OOD conditions.

---

**Weakness:**

**1. Hard to follow presentation.** The overall narrative is dominated by bullet-item style exposition and reads more like a coursework report than a polished research paper. The methodology largely lists components without sufficient intuition or connective narrative, and several key technical details are deferred to the supplementary material. In the experiments, the reporting can be difficult to interpret because some baseline outputs are excluded as “unrelated or ambiguous”, yet the paper still emphasizes “average improvement” headline numbers. Concretely, in Table 3, GENIE appears unable to perform semantic segmentation, so it is unclear why segmentation is treated as a valid task for the method; in Table 4, it is also unclear why Wang et al. (2023b) is included if it does not produce valid outputs for any task in the table.

**2. Reference-pair requirements at the inference phase are not studied.** The paper does not report sensitivity to the number of exemplars, exemplar selection strategy, or mismatch between reference and query content in the inference stage. Yet, these factors are central to the task-agnostic/generalization narrative, and could strongly impact evaluation performance. This requires systematic study and explicit reporting.

**3. “Task-agnostic” claim seems overstated.** The formulation assumes access to a reference input-output pair, which constitutes fully supervised task demonstration at inference time. A more accurate framing would be “task-agnostic architecture” or “exemplar-conditioned i2i,” rather than “task-agnostic learning” in the sense of weak supervision or no per-task paired data.

**4. Unclear and inconsistent use of “OOD”.** In the abstract, “OOD tasks” refer to deraining and contrast enhancement. Elsewhere, the evaluation includes ID tasks on ID data, OOD tasks from unseen datasets, and ID tasks on OOD domains (e.g., DomainNet). Additionally, the paper labels scribble conversion as an “OOD task” (Fig. 2 caption), which is not part of the two quantitative OOD tasks highlighted in the abstract. This terminology needs to be clarified and used consistently to enhance clarity.

**Audience:**

Yes

**Audience Explanation:**

The work proposes a practically useful idea (dual visual conditioning for exemplar-guided diffusion) and evaluates it across a broad set of i2i tasks. It should be of interest to researchers in computer vision, particularly those working on image-to-image transformation and conditional generation.

**Broader Impact Concerns:**

I have no broader impact or ethical concerns.

**Claims And Evidence:**

No

**Claims Explanation:**

In its current form, the central claims feel under-described or inconsistently presented, and several evaluation choices and missing reporting details introduce ambiguity that falls short of TMLR’s standards for clarity and careful claim-making.

**Requested Changes:**

Please see the weakness part for requested revisions. In addition, I suggest to report failure rate explicitly and define how the “average gain” is computed (e.g., average over tasks where all methods produce valid outputs vs penalizing invalid outputs).

---

> ### Author Response · Authors · 2026-01-20
> **Reference pair requirement**
>
> We thank the reviewer for highlighting the importance of exemplar selection and sensitivity analysis. We have now explicitly addressed this in the revised paper by adding a new subsection.
>
> **Exemplar selection strategy:** We would like clarify that reference input–output pairs are sampled uniformly at random during training, and no curated or retrieval-based strategy is assumed at inference. This design choice intentionally mirrors practical test-time usage, where users may provide arbitrary exemplars without guarantees of semantic similarity.
>
> **Mismatch between content of reference and query:** We also analyzed sensitivity to domain and reference-query content mismatch. As shown in Fig. 6 of the original paper, GENIE remains robust at task inference even when reference and query images differ significantly in domain or appearance, indicating that the model relies on consistency of the transformation.
>
> **Sensitivity to number of exemplars:** GENIE is inherently designed around a fixed $2 \times 2$ grid structure that encodes a single reference input–output pair and a single query. As a result, directly increasing the number of reference exemplars would require architectural changes and retraining. As an alternative that does not modify the grid structure, GENIE can incorporate multiple exemplars through explicit visual guidance. Given $K$ reference pairs, we compute individual transformation vectors $v_r^{(k)} = \varepsilon_\psi(y_r^{(k)}) - \varepsilon_\psi(x_r^{(k)})$ and aggregate them by averaging to obtain a single task representation $\bar{v_r} = \frac{1}{K}\sum_{k=1}^{K} v_r^{(k)}$. This aggregated vector is then used to form the explicit condition $c_v = \varepsilon_\psi(x_q) + \bar{v}_r$. We report the performance for different $K$ for super resolution task:
>
> | K | 1 | 2 | 3 | 4 | 5 | 6 | 7 | 8 | 9 | 10 | 11 | 12 |
> |-------|-------|-------|-------|-------|-------|-------|-------|-------|-------|-------|-------|-------|
> | PSNR | 22.323 | 22.412 | 22.452 | 22.531 | 22.582 | 22.618 | 22.643 | 22.661 | 22.671 | 22.669 | 22.671 | 22.670 |

---

> ### Author Response · Authors · 2026-01-20
> **Task-agnostic claim clarification**
>
> We would like to clarify the reviewer that our contribution lies in a task-agnostic architecture that enables task switching across multiple image-to-image transformations without any task-specific modifications to the model architecture, loss functions, or training procedure. In our setting, the model is not provided with explicit task identifiers like text prompt or any other metadata, instead it infers the intended transformation implicitly from a reference input–output examples via our conditioning mechanism. Our paper is not intended to be posed as task-agnostic learning in the sense of weak supervision or the absence of paired task demonstrations at inference time.
>
> We have revised the manuscript to more explicitly adopt the framing of exemplar-conditioned image-to-image translation and to clarify that “task-agnostic” refers specifically to the architectural design, rather than to the supervision regime. This distinction is now clearly stated to avoid potential ambiguity.

---

> ### Author Response · Authors · 2026-01-20
> **OOD terminology clarity**
>
> We thank the reviewer for pointing this out. We have updated the main paper to clarify the distinction.

---

> ### Author Response · Authors · 2026-01-20
> **Presentation improvement**
>
> We thank the reviewer for this detailed and constructive feedback. In response, we have revised the manuscript to improve the overall presentation. Specifically, reducing bullet-style exposition, added connective explanations and moved several key technical details from the supplementary material into the main text to better align the paper with a polished research format.
>
> **Segmentation clarification:** We want to clarify that while GENIE does not always produce correct class labels, it consistently segments and groups semantically coherent regions, as demonstrated qualitatively in Figs. 5, 6, and 7 of the main paper that's why it is treated as valid task. Taking into consideration one of the reviewers point, we now use a label-agnostic, clustering-based metric (Adjusted Rand Index), which captures whether the pixels belonging to the same object are grouped together, without penalizing incorrect class assignments. This provides a more meaningful assessment of structural segmentation quality when semantic labeling is imperfect.
>
> Final performance metric is as follow:
> | Metric (higher is better) | Visual Prompt | Painter | LVM  | GENIE |
> |--------------------------|---------------|---------|------|-------|
> | ARI                      | 0.303         | 0.650   | 0.573| 0.613 |
>
> We observed that GENIE achieves the second-best performance among the compared methods, trailing Painter by approximately 5%. We include this metric in the revised table for completeness and ease of comparison. In addition, we clarify in the revised manuscript that GENIE achieves an 8.30% improvement in performance on in-distribution tasks and data.
>
> **Including Painter (Wang et al. 2023b):** As Painter does not infer tasks from exemplar input–output pairs, it often produces semantically invalid outputs in the OOD task on OOD data setting, as shown in Fig. 5 of the original paper. This makes quantitative metrics misleading. This limitation is now clearly stated to improve interpretability. For this reason, we omit Painter (Wang et al. (2023b)) from the OOD quantitative comparison and will add a dedicated discussion explicitly stating this limitation in Section 4.4 (b) to improve clarity and interpretability for readers. For your reference, we are adding the performance metric of painter on the original OOD task on OOD data:
>
> | Task | Contrast Enhancement |        |        | Deraining |        |        |
> |---------|----------------------|--------|--------|-----------|--------|--------|
> |         | PSNR                 | SSIM   | LPIPS  | PSNR      | SSIM   | LPIPS  |
> | Painter | 4.127                | 0.0044 | 0.871  | 6.024     | 0.067  | 0.7426 |
>
> **Average gain clarification:** Taking the reviewers comment in account, we clarify and revise our evaluation protocol and headline metrics to ensure consistency and transparency. After incorporating a clustering-based metric for semantic segmentation, previously not used because of reason mentioned above, we recompute the average performance on in-distribution tasks and data. With this update, GENIE achieves an average improvement of **8.3%**, computed as the mean percentage change (with respect to the second best performing model in that task) across all in-distribution tasks on in-distribution data.
>
> For OOD tasks on OOD data, Painter was not included in the quantitative comparison due to its inability to produce semantically valid exemplar-based outputs therefore adding it does not affect GENIE’s relative standing with respect to other baselines, however we will add painter number in the table of the updated table. Additionally, following another reviewer’s suggestion, we introduce two new OOD tasks and report results on these settings. With these additions, the average improvement on OOD tasks and data is **5.96%**. Taking the mean across both in-distribution and OOD settings, the overall average improvement is **7.13%**, revised from the previously reported 10%.
>
> We have updated the paper accordingly to reflect these corrected and more interpretable metrics.

---

### Author Response · Authors · 2026-01-28
**Updated Paper**

This is a gentle reminder regarding our query on whether the revised manuscript must strictly adhere to the 12-page limit, or if a minor extension of 1–2 pages is permissible to fully address the reviewers’ comments.

For now, we have uploaded a revised version of the manuscript, with revisions marked in red, for the reviewers’ reference.

---

### Author Response · Authors · 2026-02-24
**Update on timeline**

Dear Action Editor and Reviewers,

I hope you are doing well. I am writing to kindly inquire about the status of our submission. According to the initial timeline, the decision was expected approximately two weeks ago, but we have not yet received an update.

We would sincerely appreciate it if you could kindly provide an update on the current status or an estimated timeline for the decision.

Thank you very much for your time and effort in handling our submission

Regards.

---

### Decision · Action_Editor_s8bC · 2026-03-27

**Recommendation:** Accept with minor revision

**Additional Comments:**

First, I want to sincerely apologize to the authors for the unacceptable delay in reaching this decision due to the previous Action Editor's unresponsiveness.

The reviewers were unanimously in favor of acceptance following the thorough rebuttal. The revised paper effectively scoped the claims, improved the interpretability of the evaluation metrics, and strengthened the OOD experiments.

Regarding the outstanding question on the page limit: an extension of 1–2 pages is perfectly acceptable and encouraged so that you can fully accommodate the reviewer-requested changes and additions in your final camera-ready version. Please ensure all promised rebuttal changes are fully integrated.

**Audience:**

Yes

**Audience Explanation:**

The paper tackles a highly relevant problem in visual-conditional image processing. Proposing a unified ControlNet-diffusion framework driven purely by visual exemplars (without the need for text prompts or auxiliary metadata) is a practically useful and elegant approach. This work will be of interest to researchers and practitioners in the TMLR community focused on computer vision, generative modeling, and image-to-image translation.

**Claims And Evidence:**

Yes

**Claims Explanation:**

While the initial submission suffered with overstated "task-agnostic" claims, misaligned semantic segmentation metrics, and insufficient quantitative evidence for out-of-distribution generalization, the rebuttal and revised manuscript successfully resolved these shortcomings. The revised version clearly re-scoped the paper claims to appropriately emphasize a task-agnostic architecture rather than a zero-shot supervision regime. Additionally, by adopting the label-agnostic Adjusted Rand Index  for a fairer evaluation of structural clustering and introducing robust, large-scale quantitative evaluations for OOD tasks like map-to-aerial translation and scribble generation, the revised paper now thoroughly grounds its core capabilities in accurate, convincing, and clear empirical evidence.